# A General Picture of Cucurbit[8]uril Host–Guest Binding: Recalibrating Bonded Interactions

**DOI:** 10.3390/molecules28073124

**Published:** 2023-03-31

**Authors:** Zhaoxi Sun, Qiaole He, Zhihao Gong, Payam Kalhor, Zhe Huai, Zhirong Liu

**Affiliations:** 1College of Chemistry and Molecular Engineering, Peking University, Beijing 100871, China; 2AI Department of Enzymaster (Ningbo) Bio-Engineering Co., Ltd., North Century Avenue 333, Ningbo 315100, China; 3School of Micro-Nano Electronics, Zhejiang University, Hangzhou 310027, China; 4Hangzhou Global Scientific and Technological Innovation Center, Zhejiang University, Hangzhou 310027, China; 5Institute of Nanotechnology, Karlsruhe Institute of Technology, Hermann-von-Helmholtz-Platz 1, 76344 Eggenstein-Leopoldshafen, Germany; 6XtalPi—AI Research Center (XARC), 7F, Tower A, Dongsheng Building, No. 8, Zhongguancun East Road, Beijing 100083, China

**Keywords:** Cucurbit[8]uril, host–guest interaction, binding mode, force fields, abused drugs

## Abstract

Atomic-level understanding of the dynamic feature of host–guest interactions remains a central challenge in supramolecular chemistry. The remarkable guest binding behavior of the Cucurbiturils family of supramolecular containers makes them promising drug carriers. Among Cucurbit[*n*]urils, Cucurbit[8]uril (CB8) has an intermediate portal size and cavity volume. It can exploit almost all host–guest recognition motifs formed by this host family. In our previous work, an extensive computational investigation of the binding of seven commonly abused and structurally diverse drugs to the CB8 host was performed, and a general dynamic binding picture of CB8-guest interactions was obtained. Further, two widely used fixed-charge models for drug-like molecules were investigated and compared in great detail, aiming at providing guidelines in choosing an appropriate charge scheme in host-guest modelling. Iterative refitting of atomic charges leads to improved binding thermodynamics and the best root-mean-squared deviation from the experimental reference is 2.6 kcal/mol. In this work, we focus on a thorough evaluation of the remaining parts of classical force fields, i.e., the bonded interactions. The widely used general Amber force fields are assessed and refitted with generalized force-matching to improve the intra-molecular conformational preference, and thus the description of inter-molecular host–guest interactions. The interaction pattern and binding thermodynamics show a significant dependence on the modelling parameters. The refitted system-specific parameter set improves the consistency of the modelling results and the experimental reference significantly. Finally, combining the previous charge-scheme comparison and the current force-field refitting, we provide general guidelines for the theoretical modelling of host–guest binding.

## 1. Introduction

Pharmaceutical research acts as a key component in biological and medical science. Great efforts are required to grasp some basic insights into druggable species from the universe of chemical space. By definition, drugs are small molecules that induce physiological or psychological changes in specific target organism(s). Despite their small sizes, they are able to regulate the functionality of huge biomacromolecular receptors upon binding to a specific area(s) of the targeted molecule or its biologically relevant counterparts [1,2,3,4,5]. The consumption of drugs should follow designed prescriptions to avoid overdosage, misuse and abuse. Otherwise, injuries in various organisms would be triggered [6,7,8,9,10,11,12,13,14].

The family of the pumpkin-like supramolecular container Cucurbit[*n*]uril (CB[*n*]) is formed by *n* glycoluril monomers linked by 2*n* methylene bridges. Their hydrophilic carbonyl portals on glycoluril units stabilize cationic species and act as hydrogen bond acceptors, while the central partly closed hydrophobic cavity is able to coordinate aliphatic and aromatic components. As a result, this series of heterocycles are able to form stable host–guest complexes with a wide range of neutral and cationic species with diverse structural features, which makes them promising coordinators and carriers of functional species including drugs [15,16,17,18,19,20,21,22,23,24]. Among the CB[*n*] family, Cucurbit[8]uril (CB8 or CB[8], see Figure 1a) has an intermediate portal size and cavity volume, which enables it to exploit almost all host–guest recognition motifs formed by this host family, e.g., both 1:1 and 1:2 host–guest binding [15,18,25]. Thus, structural and thermodynamic understandings of CB8–guest interactions would be useful to elucidate the functionality of the whole family.

In molecular simulations of condensed matter systems, especially biological systems, the high dimensional space needs to be sampled extensively in order to reach the ergodic assumption [26,27,28,29,30]. The accuracy of time integration of Hamilton equations limits the time step to several fs [31,32,33,34,35], while phenomena of interest normally happen across μs, ms and even longer time scales [36,37,38,39]. However, molecular dynamics (MD) simulations of biological molecules with modern computational resources accessible can only achieve tens of μs, which is still far from the biologically relevant time scales. The spirit of proper sampling in the conformational space, often named importance sampling, significantly extends the accessible time horizon [40,41,42,43,44]. There are many ways to improve sampling efficiency. For instance, biasing potentials can be added to the system under investigation to restrain the sampling to specific regions of phase space or enable a more flexible walk on the potential energy surface [45,46,47]. The important slow degree of freedom that the biasing potential is added onto is referred to as reaction coordinate or collective variable (CV), which is selected intuitively or according to some statistical analyses [48,49,50,51,52]. As the biasing potential modifies the energy function of the original system, a post-processing procedure is required to recover the statistics in the unbiased ensemble, which is often performed with estimators constructed from free energy perturbation. An alternative route to deal with the time-scale problem is the alchemical method, which constructs an artificial pathway connecting states of interest [53,54,55,56]. Although the method could be successful when the conformational ensemble relevant to the alchemical transformation is simple, only limited thermodynamic information could be extracted. In cases where the conformational ensemble is rather complex with many hidden barriers, enhanced sampling in the configurational space is still required. Reliable statistics could be obtained with converged sampling results, but the accuracy of the simulation results depends on another factor in molecular modelling, i.e., the Hamiltonian or model used to describe the simulated system. Although the quantum mechanics (QM) treatment is accurate and transferrable [57,58,59,60,61,62], all-atom molecular mechanics (MM) force fields are widely employed for biological systems due to efficiency and implementation (hardware/software) considerations and the size, complexity, and intrinsic time scale of the investigated problem [63,64,65,66,67,68,69].

Due to the symmetric feature and the simplicity of the CB8 structure, most modelling works of similar host–guest systems employ an a priori assumption that only a single host–guest interaction mode exists [21,70,71,72,73,74,75,76,77]. In this way, the computational modelling of host–guest interactions could be simplified to a single-state calculation, where single-point calculations with normal mode approximation or restrained sampling in the neighborhood of the initial bound conformation could be employed without the introduction of significant systematic bias. However, in our previous extensive modelling of similar host–guest systems, a complex conformational ensemble is observed [78,79]. The conformational fluctuations of the individual components involved in host–guest interaction, the orientation of specific functional groups and the diverse structural features of guests could lead to the existence of multiple unexpected binding modes, the free energy barriers between which are well above the thermal energy at physiological condition. Therefore, such a single-mode assumption is generally invalid, even for structurally simple host–guest systems.

In our previous work, the binding of seven commonly abused drugs to the CB8 host was investigated extensively [78]. The simulation does not rely on such single-state approximation but explores the configurational space extensively. The structural features of the considered drugs (G1 Metamfetamine, G2 Fentanyl, G3 Morphine, G4 Hydromorphone, G5 Ketamine, G6 Phencyclidine/PhenylCyclohexylPiperidine, and G7 Cocaine, see Figure 1a) are quite diverse, which makes it possible to conclude a general picture of the CB8–guest binding. The computational perspective generally suggests that the host tends to become squashed to hold the guest tightly, and such host–guest coordination is dynamic with many highly populated interaction modes. Although the previous work provides valuable insights into CB8–guest binding and host–guest modelling, further developments are still in sight.

All-atom force fields describe inter-molecular interactions with electrostatics and vdW terms, the former of which contributes the most significant part of non-bonded interactions. Changes in atomic charges could perturb the local interaction network and thus lead to significantly different thermodynamic profiles [80,81,82,83,84]. The polarization effect due to changes in the local environment could also play an important role in molecular recognition [85,86,87]. Two widely used charge schemes are assessed extensively for CB8–drug complexes in our previous work, and the restrained fitting method targeting the molecular electrostatic potential (ESP) is observed to provide better performance for binding affinity. To further improve the accuracy of electrostatics, polarizable models and even QM calculations are needed. Aside from the non-bonded terms that are directly involved in the calculation of inter-molecular interaction energies, the other terms describing the intra-molecular interactions are also crucial to properly describe the molecular ensemble. Different from the non-bonded interactions, these bonded terms describe the stiffer or harder degrees of freedom inside each molecule. Among the stiff degrees of freedom, the dihedral term is relatively soft, and plays a key role in the conformational flexibility of biological macromolecules and small drug-like molecules. A balanced description of intra- and inter-molecular interactions (i.e., bonded and non-bonded interactions) is required to properly describe the inter-molecular interaction pattern. Therefore, in this work, we focus on the assessment and recalibration of the intra-molecular interaction terms. The general Amber force field (GAFF) [88] and its derivatives are widely applied to describe the bonded terms in the modelling of drug-like molecules and thus also host–guest systems [89,90,91,92]. Therefore, in the current work, we assess the first and second generations of GAFF (GAFF 1.81 and 2.11, to be specified) extensively for CB8–drug complexes. A worth-noting deficiency of these force fields is that they are pre-fitted and aim at providing transferable descriptions for almost all drug-like molecules and thus could be of relatively low accuracy for specific systems. Thus, we also consider refitting the bonded terms with the generalized force-matching (FM) method [93,94] for the host and guest molecules under investigation. The resulting refitted force field is expected to provide a description closer to the target Hamiltonian (i.e., the QM Hamiltonian used to re-generate the parameters of the bonded terms) and thus more accurate than the highly transferable GAFF derivatives. Combining the central results in the current bonded-term recalibration and our previous charge-scheme assessment, some useful guidelines for the modeling of CB8 host–guest systems could be summarized. More generally, the results are expected to be useful in the modelling of all host–guest complexes.

## 2. Results and Discussion

### 2.1. Recalibrating the Bonded Interactions

Although the bonded terms are not directly involved in the calculation of inter-molecular interactions, they do have impacts on the intra-molecular conformational preference of each molecule involved in host–guest interactions. Therefore, they indirectly influence the host–guest interactions and should be properly accounted for in molecular modelling. Here, we consider refitting the bonded interactions with non-bonded terms fixed to their best-performing parameters obtained in our previous work.

Generalized force-matching refitting of the parameters in the bonded interactions is performed with the recent ParaMol implementation [94]. The reparameterization of the bonded interactions starts from GAFF or GAFF2, and every component of the atomic forces [95] and the system energy are included in the objective function. Namely, the current refitting procedure considers energy- and force-matching simultaneously. To avoid overfitting and the existence of unphysical parameters, a weak harmonic (L2) regularization term is applied to restrain the parameter space to explore in the parameter adjustment. The relative weights of the energy and force terms are the same, and that of the regularization term is 0.1 of the other two terms, leading to an intermediate level of regularization. The parameter space is further regularized according to the atom-type symmetry defined in the initial reference force field, i.e., GAFF or GAFF2. Only bonded terms including harmonic bond stretching, angle bending, and dihedral/torsion terms are refitted, and the other non-bonded terms (i.e., atomic charges and vdW terms) remain unchanged. The dihedral periodicity is also fixed in the refitting. Note that there is a special type of non-bonded interaction named the 1–4 non-bonded term. Similar to the dihedral term, this 1–4 non-bonded interaction is calculated between atoms separated by three consecutive bonds. This term is often included in the bonded terms by definition and plays an important role in correcting the errors of the dihedral/torsional term. However, as the atomic charges and vdW terms are kept fixed in our refitting, we do not touch the 1–4 non-bonded scaling constants and keep them fixed.

To construct a set of independent configurations for generating reference data, for each system, we perform 110 ns gas-phase sampling with a sampling interval of 5 ps. A high temperature of 600 K is used in the unbiased simulation to enable unconstrained exploration of the configurational space. The obtained 22,000 structures are used to initiate QM calculations. We have tested that this sample size or sampling time is already sufficient to converge the parameter set. Further adding samples (i.e., lengthening sampling time) leads to negligible changes in the outcomes. The selection of the target level in refitting determines the accuracy of the re-parameterized model. Here, we choose two target QM Hamiltonians. The first one is the computationally efficient semi-empirical QM (SQM) Hamiltonian of the dispersion, hydrogen-bonding, and halogen-bonding corrected PM6-D3H4X [96], which is shown to have good performance in the modelling of non-covalent interactions among SQM Hamiltonians. As for the second target Hamiltonian, we choose the ab initio level BLYP/def2-SVP [97,98,99,100] with the resolution-of-the-identity approximation [101,102] and the latest D4 dispersion correction [103,104] considering its good performance in modelling various properties including non-covalent interactions among pure functionals [105]. Semi-empirical calculations are performed with the MOPAC2016 program [106,107], while for ab initio calculations the ORCA software [108,109] is employed. Note that the configuration sets for PM6-D3H4X and BLYP-D4/def2-SVP are generated independently. Thus, the parameters and errors of the refitted models are not correlated.

We first discuss the PM6-D3H4X results. To assess the quality of the refitted parameters, we generate another 25 ns trajectory (5000 configurations) at two temperatures with the refitted parameter set FM-PM6. The first temperature is the one that we generate in the configurational ensemble in refitting, namely 600 K. This high-temperature set enables the accuracy assessment in the ensemble in that the refitting is performed. A lower temperature of 300 K is also employed to generate another set of configurations, which enables the assessment of energetics in the low-energy regions accessible at room temperature. Two error metrics including the root-mean-squared error (RMSE) and mean absolute error (MAE) are used for quality assessment.

In quality assessment, we first take a detailed investigation of the parameter set for the host CB8 due to its importance in coordinating all guests. Although (S)QM Hamiltonians are more accurate than MM force fields, all of them have their own weakness in specific regions/areas. Therefore, we then cross-check the refitted parameter sets and test whether the force field refitted targeting PM6-D3H4X is able to reproduce the energetics and atomic forces at another QM level. Here, we choose another SQM Hamiltonian of RM1 [110]. A similar sampling procedure is performed (25 ns gas-phase sampling at 300 K for 5000 configurations). The correlations between the MM and SQM energetics when using PM6-D3H4X and RM1 as references are provided in Figure 2a,b, respectively. As the refitting is performed using the PM6-D3H4X data, the errors are smaller when the same SQM Hamiltonian is used as the reference level in the assessment, as shown in Figure 2a. When the reference level in quality assessment is set to another SQM level (i.e., RM1 here), interestingly, the errors of energetics are still smaller than the original GAFF2 parameters, as shown in Figure 2b. However, the error produced by the refitted force field (i.e., FM-PM6) is obviously larger than in the previous case (c.f. Figure 2a). These observations suggest that the refitted force field targeting PM6-D3H4X produces descriptions closer to a series of (S)QM Hamiltonians, not just its target level. The time series of atom-specific force errors (Frobenius norm) with the original GAFF2 and the newly fitted parameter set is calculated for the PM6-D3H4X reference in Figure 2c,d, respectively. We can see that during the entire 25 ns sampling, the force errors are very large with the original GAFF2, and significant improvements are observed upon force-field refitting. The RMSE of the atomic forces is lowered from 40.05 kcal/(mol·Å·atom) to 17.87 kcal/(mol·Å·atom). To obtain further insights into the error distributions among atoms, the heavy atoms are sorted to the first 96 indices, and the others are hydrogen atoms. In this way, from Figure 2c,d, we can see that the force errors are significantly larger for non-hydrogen atoms, which suggests that atomic forces on heavy atoms could be more problematic than on hydrogen atoms. As the conformational preference of a molecule is directly related to the rearrangement of torsional degrees of freedom involving these non-hydrogen atoms, a solution to further improving the fitting quality for atomic forces on these heavy atoms is attaching more importance to the force terms on these heavy atoms in the objective function, i.e., using larger weights for these terms. In this way, the force errors on these non-hydrogen atoms would be lowered and the conformational preference at the target level would be reproduced more accurately. Note that the force errors when selecting the RM1 Hamiltonian as the reference level have similar behaviors (results not shown). Namely, the force errors produced by the refitted force field are smaller than the GAFF2 results, although the reference RM1 level is not its fitting target. The observation about force errors agrees with the improvement of the system energy in Figure 2b and suggests that the refitted force field produces better descriptions of many properties closer to a series of (S)QM Hamiltonians.

The better reproduction of PM6-D3H4X energetics for the host CB8 under the refitted force fields is also valid for other molecules (i.e., guests), the energetics correlation plots of which are provided in Appendix A. We can see that for all molecules the refitted force field produces energies closer to the y = x line and thus smaller deviations from the reference level, which suggests that the refitted force field provides more accurate descriptions compared with the original GAFF2 parameter set. The time series of the force errors under GAFF2 and the refitted force field for all molecules are presented in Appendix A. We can see that for all molecules the refitted parameter set provides smaller force errors for all atoms during the entire 25 ns unbiased sampling. The accuracy improvement of the refitted force field is observed at both temperatures, which indicates that the refitted force field reproduces the target PM6-D3H4X results more accurately in both the high- and low-energy regions.

The comparison between RMSEs of the energetics produced by the original GAFF2 and the newly fitted parameter set with configurations sampled at 300 K is provided in Figure 2e, where smaller deviations from the PM6-D3H4X reference are observed for all molecules under the refitted parameter set FM-PM6. The RMSEs of atomic forces under the two force fields are shown in Figure 2f, where the overall deviations of the forces are smaller under the refitted force field. Therefore, the FM-PM6 description is closer to the target Hamiltonian PM6-D3H4X than the original GAFF2 parameter set. It is worth noting that we have also investigated the impact of the regularization strength on the fitting outcome to avoid over- and under-regularized fitting. The regularization term used in the current refitting generally leads to an error increase of about 10%, and the obtained parameters are within the reasonable range of physical properties. Overall, the refitted force field FM-PM6 provides a description much closer to the target level PM6-D3H4X than the original GAFF2 in all relevant configurational regions. Then, we turn to the other refitting target BLYP-D4/def2-SVP.

Similar procedures are repeated for the other target BLYP-D4. Namely, for each molecule, we perform 110 ns unbiased sampling at 600 K in vacuo to obtain 22,000 configurations for ab initio calculations in parameterization. As for the quality assessment, as the computational cost is much higher under the ab initio level, the sampling interval is increased to 10 ps and the number of samples is set to 1000. As a result, the sampling time for each molecule becomes 10 ns in quality assessment. The validation test (i.e., sampling and QM calculations) is only performed at 300 K.

Still, we first look at the CB8 results due to its involvement in all host–guest pairs. Two reference QM levels including the target BLYP-D4 and another QM level PM6-D3H4X (i.e., the previous target in the FM-PM6 refitting) are used in this assessment. The configurations at the two levels are generated independently and thus the results are uncorrelated. The energetics comparisons are shown in Figure 3a,b. When the fitting target BLYP-D4 is selected as the reference level, as shown in Figure 3a, the refitted force field provides much smaller errors for energetics than the original GAFF2. Interestingly, when the previous target PM6-D3H4X is selected as the reference level, the FM-BLYP parameter set still provides smaller errors than GAFF2, which indicates that the refitted force field provides descriptions not only closer to its target level but also similar to some SQM Hamiltonians. This phenomenon also suggests that the semi-empirical level PM6-D3H4X is of good quality. Another interesting comparison is between the two quality assessments in Figure 2a and Figure 3b. In both cases, the reference Hamiltonian is the PM6-D3H4X, but the parameter sets under assessment are FM-PM6 and FM-BLYP, respectively. The first FM-PM6 parameter set is fitted targeting the reference level, while the second FM-BLYP is not. This leads to higher errors produced by FM-BLYP. Interestingly, the GAFF2 errors (RMSE and MAE) in the two assessments are very similar, but the latter assessment only uses 1000 configurations (1/5 of the former), which suggests that the quality assessment already converges or is bias-free at this sample size. Therefore, the 1000-configuration assessment in the FM-BLYP case is reasonable and reliable. The time series of the by-host-atom decomposition of the force errors are presented in Figure 3c,d for the original GAFF2 and the refitted parameter set, respectively. The reference level is selected as the fitting target BLYP-D4/def2-SVP. It is clearly shown that during the entire 10 ns unbiased sampling, the force error on each atom is quite large with the original GAFF2, and the situation is improved significantly upon refitting. The RMSE of force errors decreases from the GAFF2 30.06 kcal/(mol·Å·atom) to the FM-BLYP 13.07 kcal/(mol·Å·atom). Similar to the previous FM-PM6 case, the force errors are larger for heavy atoms (i.e., No. 1–96 atoms), which is expected as the limited number of degrees of freedom coupled to light atoms (hydrogen). The statistics of the guest molecules are given in Appendix A (energetics) and Appendix A (${\left\|\Delta F_{i}\right\|}_{2}$), where similar phenomena (i.e., better reproduction under the refitted force fields) could be observed.

The RMSEs of energetics produced by GAFF2 and FM-BLYP are shown in Figure 3e, where the refitted force field produces much smaller energy deviations for all molecules under consideration. The situation of atomic forces is similar. As shown in Figure 3f, the force errors are much smaller under the refitted force field, compared with the original GAFF2. Overall, as long as the refitting procedure is performed, the newly fitted parameter set produces descriptions closer to the target level. As the new parameter sets are constructed in a molecule-specific manner and the targets BLYP-D4 and PM6-D3H4X are of good accuracy, they are expected to provide more accurate results than the initial guess GAFF derivatives. Note that the force field refitting is also performed with another initial guess of GAFF for the PM6-D3H4X target. As the fitting procedure and quality assessment are similar to the presented cases, we would not discuss them here.

### 2.2. A Closer View of Different Parameter Sets

The comparisons for energetics and atomic forces presented above show that the refitted parameter sets reproduce the target Hamiltonians in a better way than the pre-fitted parameter sets. However, the detailed differences between different force fields remain unclear. Therefore, before performing μs-length enhanced sampling to explore the configurational space of host–guest complexes, we first assess these parameter sets in detail to understand their intrinsic behaviors and the differences between them. The host CB8 is still selected as the illustrative example. The difference between vdW parameters in the two GAFF force fields is small. Thus, we focus on the bonded terms.

The torsional rearrangement is the most important influencing part for conformational preferences in bonded interactions, while the other terms (i.e., bond stretching and angle bending) are relatively irrelevant. Therefore, we investigate the torsional/dihedral term in each force field in detail. The first comparison to look at is between the GAFF derivatives. Dihedral terms defined in GAFF derivatives that are used to describe the conformational preference (stiffness) of the host CB8 are shown in Appendix A. The parameters of most dihedral terms in GAFF and GAFF2 for the host CB8 are similar (some are just identical), and the main difference between the GAFF and GAFF2 parameter sets lies in the torsional barrier term of the N-C-N-C dihedral presented in Figure 4a. Specifically, there is no torsional barrier explicitly defined in GAFF, while that in GAFF2 is 2.08 kcal/mol. Namely, the stiffness of the CB8 ring in this region is maintained by the other dihedral terms shown in Appendix A in GAFF, while in GAFF2 an explicit torsional barrier is added. The phase of this dihedral is set as zero, which suggests that the addition of the torsional barrier strengthens the stiffness of the CB8 ring. Therefore, GAFF2 is expected to provide a stiffer CB8 ring than its precedent.

To provide further details of the dynamics of the bonded parameter sets of the CB8 host, we then perform unbiased MD simulations in explicit solvent for 20 ns with a sampling interval of 5 ps under GAFF derivatives. From the structural overlay in Figure 4b,c, we can see that the CB8 ring sometimes visits the squashed conformation under GAFF even in the absence of the external guest, while under GAFF2 the host ring is quite stiff and the fluctuations of the ring are very small. These observations suggest that the GAFF2 CB8 is stiffer than the GAFF one, which is in perfect agreement with the above investigation of the detailed definitions of the torsional potential. As for the refitted parameter sets, as all bonded terms are adjusted under regularized optimization, it is difficult to identify which term has the largest contribution to the different behaviors of different parameter sets. Therefore, we focus on the host dynamics produced by these refitted force fields. A total of 20 ns unbiased simulations under two newly fitted models targeting PM6-D3H4X initiated from GAFF and GAFF2 are performed in order to investigate the influence of the initial guess in the regularized parameter adjustment. By comparing the FM-PM6 parameter sets initiated from GAFF and GAFF2 in Figure 4d,e, we know that regardless of the original parameter set used in regularization (i.e., GAFF and GAFF2), the refitted force fields provide flexible host rings. However, the shapes of the host ring produced by the refitted parameter sets from GAFF and GAFF2 differ. For the FM-PM6 parameter set restrained around GAFF, the host cavity seems like a cycle. By contrast, a rectangular cavity is maintained by the FM-PM6 parameters restrained around GAFF2. The RMSEs of energy and atomic forces of the FM-PM6-from-GAFF and FM-PM6-from-GAFF2 parameter sets are similar (results for the former not shown), which suggests that the two parameter sets provide descriptions with similar sizes of errors/deviations from the target level. Thus, it is difficult to say which force field is more accurate. As the GAFF force field has been employed in extensive sampling simulations in our previous work [78], in the current work we use the GAFF2 parameter set in simulations and as the initial guess used to initiate the regularized parameter optimization. Generally speaking, the BLYP-D4 level is more accurate than PM6-D3H4X and the refitted parameter set targeting BLYP-D4 is of higher reliability. Moreover, considering the system-specific fitting parameters and the resulting broken transferability, the refitted FM-BLYP parameter set is expected to be the best (most accurate) description employed in the current work. The superposed snapshots obtained under FM-BLYP-D4 from GAFF2 are presented in Figure 4f, where a different behavior of the CB8 ring is observed. The refitted FM-BLYP parameter set provides a stiff host ring similar to the GAFF2 description. As this target level is more accurate than PM6-D3H4X, this stiffer behavior of the host cavity is considered to be more reliable. Based on the above observations of the host ring in plain explicit-solvent simulations, we expect the CB8 ring to be stiff and maintain its perfect-circle conformation in the absence of the guest. On this aspect, GAFF2 and FM-BLYP perform well, FM-PM6 is of intermediate quality, and GAFF performs worst. However, we cannot guarantee that this accuracy rank is applicable to host–guest complexes in a solvent, where complex inter-molecular interactions are involved. According to the observations about the CB8–drug systems in our previous work based on GAFF [78], upon host–guest binding and the formation of strong inter-molecular interactions, the host ring could possibly be distorted and squashed to hold the guest tightly. In that case, a too stiff description of the host ring could trigger the inability of the host to squash and fit the shape of the binding guest in the bound state, leading to inappropriately looser host–guest coordination.

The radius of gyration (Rg) is often chosen as the CV to describe the open and close of the host cavity. However, the behavior of this CV is generally not investigated in detail, and thus the correctness of this selection is not justified. To analyze the detailed behavior of this CV, we compare the time series of Rg of the host ring under GAFF2 and the refitted force fields in Figure 4g. The GAFF2 curve is quite stable, which is in agreement with the host stiffness produced by this parameter set. By contrast, the Rg curves for both FM-PM6 force fields experience significant fluctuations, which suggests the higher flexibility of the host ring and the existence of an ensemble of host conformation. It is worth noting that the maximum and minimum values for the two FM-PM6 force fields are similar, and their fluctuation behaviors also show similar trends. However, the detailed structural features produced by the two refitted force fields differ, as shown in the superposed snapshots (c.f., Figure 4e,f). Therefore, this CV cannot properly differentiate different structural features produced by different force fields and thus may not be a good CV to analyze the detailed host conformation. The FM-BLYP parameter set provides a stable Rg curve and higher values of Rg than the original GAFF2 parameter set, which suggests that the host cavity described with this parameter set is wider than GAFF2. Overall, FM-BLYP and GAFF2 behave similarly and describe a stiff host ring with a wide central cavity, while the host rings described by FM-PM6 and GAFF are of higher flexibility.

### 2.3. Binding Modes

The general picture of CB8–guest interactions observed in our previous extensive modelling is a squashed CB8 ring that holds the guest tightly [78]. Intuitively, such a squashed host ring is energetically unfavorable under the stiff GAFF2 parameter set, and thus the GAFF2 binding modes are expected to show some differences compared with the GAFF ones observed in our previous work. The re-parameterized set FM-PM6 provides a host cavity with intermediate levels of flexibility and thus is expected to have different behaviors compared with the GAFF derivatives. As the host behavior under the FM-BLYP parameter set is similar to the initial guess GAFF2, this refitted parameter set is expected to have similar behaviors to GAFF2. Aside from the structural features of the host cavity, the behaviors of the guests are also altered when changing the bonded parameter set. The alterations of the host and the guest would lead to different intra- and inter-molecular interaction patterns. In the following paragraphs, we would investigate whether the previously observed general picture of CB8 host–guest binding is still valid when different bonded parameter sets are employed.

In the analysis of the host–guest coordination, we still start by monitoring the total host–guest contacts and its by-host-atom decomposition to check whether the space of the relative host–guest position is properly explored. The illustrative example chosen here is the complex formed by CB8 and the deprotonated G5 described by the FM-PM6 parameter set, as shown in Figure 5. Note that all atoms of the host and the guest are included in this calculation, while in the free energy analysis shown later only heavy atoms are considered. In Figure 5a, during the 1000 ns enhanced sampling simulation, the total host–guest contact number experiences significant fluctuations, which suggests that the exploration of the space of the relative host–guest position is accelerated. The total contact number reaches its maximum of around 1800 and approaches zero many times, which indicates that the bound and unbound states are repeatedly visited. More details about the host–guest coordination could be obtained from the by-atom decomposition shown in Figure 5b. For instance, in the cyan oval, the atom–guest contacts are larger than 1 for all host atoms, which indicates that all host atoms are in direct contact with the guest. The total host–guest contact number in this region is close to the maximum of the observed values. These phenomena suggest a typical center-binding pose, where the whole guest stays at the center of the host cavity. Another example is presented in the magenta oval. The total contact number in this region is obviously lower than the maximum value but is much larger than zero. This intermediate value suggests a looser host–guest coordination. More detailed coordination patterns could be extracted from the atom–guest contacts, where only some host atoms have non-zero atom–guest contacts. Therefore, the host–guest coordination picture in the magenta oval is that the guest binds to the outer surface of the host ring, which corresponds to a side binding pose. Similar observations could be obtained from the time series for all host–guest pairs under GAFF2, FM-PM6 and FM-BLYP presented in Appendix A. Therefore, the sampling does explore the space of host–guest coordination effectively. We then turn to the thermodynamic landscapes to investigate the stability of these explored interaction patterns.

Similar to our previous work, we still start by checking the radius-contact (ρ−C) surfaces for all host–guest pairs. The results under the pre-fitted GAFF2, the refitted FM-PM6 and FM-BLYP parameter sets are presented in Appendix A. Generally, there are one to three minima on the radius-contact surface. Some of them are very wide and do not differentiate different host–guest binding modes in a satisfactory manner. For detailed discussions please refer to Appendix A.

As the combination of the COM distance ρ and the host–guest contacts $C_{host\text{–}guest}$ cannot properly differentiate different interaction patterns under all Hamiltonians, we thus consider the $C_{host\text{–}guest}-C_{aromatic}$ or $C-C_{Ph}$ surfaces, as performed in our previous work [78]. There is at least one aromatic ring formed by six carbon atoms in each host, and the contact number between the aromatic 6-membered ring of the guest and the host is used to decompose the overall host–guest coordination into the host-aromatic and the other ones. We still only consider the heavy atoms in this free energy analysis. As there are two phenyl rings for Fentanyl G2, we select the phenyl ring directly connected to the amide bond. As the current analyses of host–guest binding patterns under different bonded parameter sets are strongly related to the discussions about GAFF binding poses reported in our previous work [78], we recommend that readers refer to the previous paper for a thorough understanding of the current results and discussions.

In the current $C-C_{Ph}$ analysis, we first check the GAFF2 results shown in Figure 6. For the structurally simplest guest under study, G1, a single wide free energy minimum is observed on the free energy surface. Structures extracted there share the center-binding feature with inter-molecular hydrogen bonds formed between -NH_2_^+^ of the guest and the -C=O portals of the host. The host cavity is less distorted compared with the GAFF case, which is related to the higher stiffness of the GAFF2 description. The free energy surface for the structurally complex G2 shows obvious differences compared with the GAFF result. Under GAFF2, the binding poses with large host-phenyl contacts are no longer stable, and only conformational states with small $C_{Ph}$ are favorable. On the free energy surface, the most stable binding pose lies in the state with the smallest host–phenyl contacts. In this conformational state, the phenyl ring on the -NH^+^ side penetrates the cavity, the amide bond part of the guest clings to one side of the host ring, and the phenyl connected to the amide bond stays away from the host. The inter-molecular -NH^+^ ··· -C=O distance is longer than the normal criterion for hydrogen bonds, suggesting the absence of host–guest hydrogen bonds. It is worth noting that compared with the GAFF case where the host ring is squashed significantly in host–guest coordination, under GAFF2 the host ring is perturbed at a minimal level, which is expected due to the higher host flexibility defined by the GAFF parameters discussed extensively in the previous section. The minimum next to the first minimum has larger host–guest and host-phenyl contacts. This conformation is of the highest stability under GAFF, but under GAFF2 it is about 0.5 kcal/mol less stable than the previous minimum. In this binding pose, the guest nestles against the host with the phenyl moiety connected to the amide bond binding to the outer surface of the host, while the other phenyl ring stays at the center of the host. Still, no inter-molecular -NH^+^ ··· -C=O hydrogen bond is formed. The differences between the GAFF and GAFF2 results suggest that different behaviors of intra-molecular terms could lead to different inter-molecular binding thermodynamics and interaction patterns in host–guest complexes. The CB8-G3 surface under GAFF2 is rather simple. For this structurally rigid guest, the only bound conformation features the whole guest inside the central cavity and is similar to the GAFF case. The -NH^+^ and two -OH groups of the guest form hydrogen bonds with the -C=O portals of the host. Although G4 is very similar to G3, the free energy surface of CB8-G4 shows more complex behaviors. The typical center-binding conformation is no longer the dominant binding pose in the host–guest coordination, but there are a series of stable binding patterns even when the inter-molecular coordination is not very compact. There are two states with obviously larger host–guest and host–phenyl contacts than the other states. These two large-contact states belong to variations of the center-binding poses, while the other could be grouped into intermediate states along the binding/unbinding pathway. The most stable bound state has the largest C but an intermediate value of $C_{Ph}$. In this state, the -OH group of the guest forms a hydrogen bond with a -C=O portal, but the -NH^+^ group is perpendicular to the host ring. The aromatic ring of the guest is relatively far from the host center compared with the other large-C pose, which is expected considering its smaller $C_{Ph}$. Compared with the global minimum, the other center-binding pose has similar host–guest contacts but larger host–aromatic (phenyl) contacts. Thus, we expect that in this pose the guest still stays at the host center, but its aromatic ring would be closer to the host center. Visualizing the structure extracted there, we confirm that the guest is indeed inside the host cavity, but its orientation differs from the first pose. The aromatic moiety of the guest is encapsulated at the center of the host cavity while the other parts of the guest are relatively solvent-exposed. When the host–guest and host–phenyl contacts begin to fall, the guest starts to leave the host center. The first free energy basin on this association/dissociation pathway next to the global minimum features a larger distance between the aromatic ring and the host center and the inter-molecular hydrogen bonds formed between -OH and -C=O portals. When the contact numbers drop further, two host–guest coordination patterns are reached. In the conformational state with larger C but smaller $C_{Ph}$, the guest becomes farther from the host center and most parts of the guest are in a solvent. Still, this binding pose has basic center-binding features and can be considered a pre-formed center-binding pose. However, the other pose with smaller C but larger $C_{Ph}$ has a totally different inter-molecular coordination pattern. At this position, the guest binds to the outer surface of the squashed host, which makes it a side-binding pose that is very unstable under GAFF. Overall, for the CB8–G4 complex, the GAFF2 description produces a variety of host–guest coordination patterns instead of the single center-binding preference under GAFF. The protonated and deprotonated G5 under GAFF2 show similar behaviors to the GAFF cases. The crescent free energy landscapes depicting the conformational fluctuations of the guest inside the host cavity are still observable, and minor differences between the GAFF and GAFF2 results lie in the detailed distributions of the low-(free-)energy regions. The protonated G5 has a narrower low-(free-)energy minimum than the deprotonated form. The bound structure extracted at the center of this minimum has the typical center-binding features, which is very similar to the GAFF result. As for the deprotonated G5, its free energy surface is relatively rugged and the low-(free-)energy minima are distributed sparsely. We extract four structures on the surface to analyze the detailed host–guest coordination patterns. The binding pose at the center of these minima is the typical center-binding one, where the whole guest is inside the host cavity. The other two minima on the crescent and in the neighborhood of the global minimum are related to the conformational fluctuations of the guest inside the host cavity. These binding poses are produced when one of the two 6-membered rings of the guest leaves the host center. The above observations about the protonated and deprotonated forms of G5 are similar to the GAFF results, but there is one new stable binding pose for the GAFF2 deprotonated G5. In the pose next to the typical center-binding position and with larger host–guest and host–phenyl contacts, the host ring becomes more squashed and the host–guest coordination becomes tighter than the other conformational states. The -NH group of the guest forms one hydrogen bond with a -C=O portal. This binding conformation is observed under GAFF but is not as stable as the typical minimum. However, under GAFF2 this binding pose is of similar thermodynamic stability to the other. This phenomenon suggests that the conformational fluctuation picture is also dependent on the bonded parameters. The structurally complex G6 also produces significantly different host–guest coordination patterns under the two GAFF derivatives. Under GAFF, two well-separated minima are observed, and the host–guest coordination is considered to be dominated by two interaction patterns [78]. By contrast, an extremely wide free energy basin containing a series of bound conformations is observed under GAFF2. This phenomenon is somehow similar to the G5 case, where a wide crescent free energy landscape that is highly related to the conformational fluctuation of the guest inside the host cavity is observed. As a result, the CB8–G6 coordination is considered to be similar to the G5 one. Namely, the host–guest coordination is highly dynamic and the conformational fluctuation of G6 inside the host cavity is significant. To understand the details of the host–guest binding pattern, we basically extract three representative structures from three distant positions on the free energy surface. The most stable binding pose or the global minimum lies at the center of this wide free energy basin. The structure at this position features the phenyl ring outside and the other two 6-membered rings inside the host cavity. The -NH^+^ group of the guest is perpendicular to the host ring, and thus no inter-molecular hydrogen bond is formed. The other two thermodynamically less stable poses are extracted from the tails of the wide free energy basin. In the pose with larger host–guest and host–phenyl contacts, the piperidinium and phenyl moieties penetrate the host cavity and the cyclohexyl ring is solvent-exposed. -NH^+^ is still almost perpendicular to the host ring, which makes the formation of intermolecular hydrogen bonds impossible. In the other one with smaller host–guest and host–phenyl contacts, both the phenyl and cyclohexyl moieties flip out and only the piperidinium ring stays inside the host cavity. From the above structural information, we can summarize the basic feature of the conformational fluctuation inside this wide free energy basin. The common feature of these three binding structures is that the piperidinium ring is always inside the host cavity or close to the host center, while one of the other two 6-membered moieties (i.e., phenyl and cyclohexyl) of both of them can flip outside the host cavity without causing significant unfavorable interactions. Another worth-noting point observed on the free energy surface is that the wide free energy minimum under GAFF2 covers almost all energetically favorable regions under GAFF, which suggests that the GAFF binding poses are a subset of the GAFF2 one. This conclusion can also be reached by comparing the host–guest binding patterns reported in our previous work and the current GAFF2 results. The last guest under investigation, G7 Cocaine, is also found to have different behaviors under the two GAFF derivatives. Unlike the GAFF case where only a single bound conformation is preferred, under GAFF2 the CB8-G7 complex has a very complex thermodynamic landscape with multiple low-(free-)energy minima. There are basically four conformational states on the $C-C_{Ph}$ surface. The most obvious difference between these states that can be identified on the free energy surface is $C_{Ph}$. The global minimum or the most stable binding pose has the largest host–phenyl contacts, which suggests a tight host–phenyl coordination at this position. In this state, only the phenyl ring is penetrating the squashed host cavity, while the other parts are far from the host center and solvent-exposed. No inter-molecular hydrogen bond is formed between the host and the guest. Note that this conformational state is also the most stable one under GAFF, and the detailed host–guest interaction patterns at this position are similar under the two GAFF derivatives. Thus, the most stable binding poses are similar under the GAFF derivatives. However, when $C_{Ph}$ drops down and the system leaves the minimum, the thermodynamic stability of the host–guest complex decreases dramatically under GAFF, while under GAFF2 a series of (meta-)stable conformations with relatively high stabilities are reached, which depicts a multi-state picture for host–guest coordination instead of the single-pose-dominant one under GAFF. In the smaller-$C_{Ph}$ state next to the global minimum, the phenyl ring inserts deeper into the host cavity and comes out from the other side, and the other parts of the guest are thus closer to the host center, which leads to smaller host-phenyl contacts and larger host–guest contacts. The existence of this binding pose is obviously due to the conformational fluctuations of the guest inside the host cavity. Note that this binding pose is observed under GAFF, but its thermodynamic stability is much lower than the first pose in that case. By contrast, this conformational-fluctuation-induced binding pose is of similar thermodynamic stabilities to the first binding pose under GAFF2. Thus, it is obvious that the conformational preferences produced by the GAFF derivatives are somehow different. When $C_{Ph}$ drops further, the phenyl moiety becomes further away from the host center. The low-(free-)energy pathway on the free energy surface for this process suggests an increase in host–guest contacts, which indicates that the other parts of the guest become closer to the host center. The structure visualization confirms this host–guest coordination picture. We can see that only the phenyl moiety is outside the host cavity, while all the other parts of the guest stay at the center of the host ring. Obviously, this binding pose is also highly related to the previous two poses through conformational fluctuations. However, unlike the previous two poses, the free energy barrier between this binding pose and the other two is high, which suggests that the interconversion between them is not easy. The 4th pose has smaller host–guest and host–phenyl contacts compared with the 3rd, which draws a picture that all parts of the guest start to leave the host center. The structure extracted here shows that the phenyl ring becomes further away from the host cavity, and the other parts of the guest also leave the host center. This seems like an intermediate when the host–guest complex starts unbinding. The above GAFF2 $C-C_{Ph}$ surfaces show obvious differences from the GAFF results in our previous work in many aspects, which suggests that the inter-molecular coordination pattern is strongly dependent on the inter-molecular interaction terms. For the host cavity, the GAFF2 parameter set differs from GAFF mainly in the stiffness of the host ring, while the parameters for the guest molecules are also changed when we shift from GAFF to GAFF2. These factors, as a whole, trigger the somehow different CB8–drug binding patterns, but the general picture of the guest dynamically coordinated by a distorted host with significant conformational fluctuations remains unchanged.

We then turn to the molecule-specific FM-PM6 parameter set. It has been observed in unbiased simulations that compared with the initial guess GAFF2, this refitted parameter set produces a more flexible host cavity. Thus, we expect to observe a flexible host in the bound state. Aside from the host behavior, as the guest descriptions are also altered upon force-field refitting, the host–guest coordination is expected to be changed also in many other aspects. The $C-C_{Ph}$ free energy surfaces under this parameter set are presented in Figure 7, and the visualization of structures is used to aid the understanding of the detailed inter-molecular coordination pattern. For the structurally simple guest G1, we observe a single free energy minimum that is much wider than the GAFF2 case. To understand the structural features in this wide conformational state, we extract three structures from the center of this minimum and its large-contacts and small-contacts tails. The system is of the highest stability when the two contact numbers are large, and the structure on the right depicts a coordination pattern that the whole guest penetrates the host cavity, the host ring becomes squashed to maximize the host–guest coordination, and the -NH_2_^+^ group of the guest forms inter-molecular hydrogen bonds with the -C=O portals of the host. In the binding pose, in the middle which is about 1 kcal/mol less stable, the guest still stays in the host cavity but is more extended than in the previous pose. The host is squashed but the host–guest coordination is not as tight as in the previous pose. Inter-molecular hydrogen bonds are formed between the -NH_2_^+^ group of the guest and the -C=O portals of the host. The structure at the left has smaller host–guest and host-phenyl contacts than the other two, which suggests a looser level of inter-molecular coordination and a larger host–guest distance. As expected, in the left structure we observe that the phenyl ring of the guest still penetrates the host cavity, but the other parts become relatively far from the host center. The inter-molecular hydrogen bonds are still formed. Overall, for the CB8–G1 binding under FM-PM6, the host cavity in the bound state is more squashed and irregularly shaped than the GAFF2 case, which is in agreement with the increased flexibility observed in unbiased simulations. As for the structurally complex G2, the FM-PM6 surface also shows obvious differences compared with the initial guess GAFF2. Generally, there are four conformational states observed on the $C-C_{Ph}$ surface, which is more similar to the GAFF case instead of the initial guess GAFF2. This phenomenon is in agreement with the flexibility of the host cavity observed in unbiased simulations under this newly fitted parameter set, which is more similar to GAFF than GAFF2. The positions of the observed four conformational states are also very similar to those observed under GAFF. However, by visualizing the structural details of the bound conformations, we find that the FM-PM6 binding poses show obvious differences from the GAFF ones. The four conformational states could be divided by their host–phenyl contacts. The two minima on the top-right have larger host-phenyl contacts, and their difference lies in the host–guest contacts, which suggests that the phenyl ring is in close contact with the host ring while the other parts of the guest fluctuate and produce different conformational features. In the state with the larger host–guest contacts, a very typical center-binding pose is observed. The whole guest is in the CB8 cavity, and the guest is folded in a compact form to insert tightly into the host cavity. The -NH^+^ group of the guest forms a hydrogen bond with one of the -C=O portals of the host. In the other state with smaller host–guest contacts, the whole guest clings to the host, and phenyl at the amide bond side penetrates the host. No inter-molecular hydrogen bond is formed. This conformational state is less stable than the first one, which is in accordance with the GAFF results. Note that these two large-$C_{Ph}$ states are thermodynamically unstable under the initial guess GAFF2, but are favorable under the more flexible GAFF. However, under either GAFF or GAFF2, the host cavity is not as twisted as the current FM-PM6 case when coordinating the penetrating groups. The other two states with smaller host–phenyl contacts also have smaller host–guest contacts and their difference mainly lies in the host–phenyl contacts. In the state with larger host–phenyl contacts, the basic picture of host–guest coordination under FM-PM6 is still similar to those observed under the GAFF derivatives. Specifically, the guest nestles against the host with the phenyl connected to the amide bond binding to the outer surface of the host, while the other phenyl ring stays at the center of the host. However, it is observed again that under either GAFF or GAFF2, the host cavity is not so twisted as in the current FM-PM6 case. In the last state with the smallest host–guest and host–phenyl contacts on the free energy surface, the basic features of this binding pose are also similar to the results under the GAFF derivatives. Namely, the phenyl ring on the -NH^+^ side penetrates the cavity, the middle part of the guest clings to one side of the host ring and the selected phenyl ring stays away from the host. The inter-molecular -NH^+^ ··· -C=O hydrogen bond seems to be formed. However, the host ring is more distorted than the GAFF and GAFF2 cases. Overall, during the CB8–G2 interactions, the basic features of the interaction patterns remain unchanged, but the host cavity is more twisted than the previous GAFF and GAFF2 cases, which is in agreement with the observations in the CB8–G1 complex. As for the structurally rigid G3, only one conformational state is observed on the $C-C_{Ph}$ surface, which agrees well with the GAFF derivatives. This center-binding pose features the whole guest inside the central cavity with its -NH^+^ and two -OH groups forming hydrogen bonds with the -C=O portals of the host. However, the host ring is more twisted and irregularly shaped compared with the other parameter sets, which is expected considering the dynamic behavior of the CB8 ring in unbiased simulations reported in the previous section. The guest G4 is structurally similar to G3, but this time under FM-PM6 it still has an obviously different $C-C_{Ph}$ surface compared with G3. The center-binding minimum with large host–guest and host–phenyl contacts is wider than the GAFF or GAFF2 case, which suggests a higher level of conformational fluctuations. Two microstates could be identified in this large-C (or equivalently large-$C_{Ph}$) basin. The most stable binding pose lies in the minimum with the largest host–guest contacts. It depicts a typical center-binding pose with the inter-molecular host–guest hydrogen bonds formed and the aromatic ring of the guest staying at the host center. The other microstate in this region shares some structural features with the first one, and their difference lies in the dynamic fluctuations of the two components involved in the inter-molecular coordination. When the host–phenyl contacts begin to fall, the phenyl ring of the guest leaves the host center and the dissociation starts. A long low-(free-)energy pathway is observed on the free energy surface. The first structure extracted on this association/dissociation pathway has larger host–guest and host–phenyl contacts and thus features tighter host–guest coordination. This binding pose has similar host–guest contacts to the second pose but its host–aromatic contact number is much smaller. Thus, in this position, the guest still stays at the host center, but its aromatic ring is more solvent-exposed (i.e., far from the host center). It is worth noting that unlike the GAFF2 case where the partly dissociated structure is of similar thermodynamic stability to the center-binding pose, under FM-PM6 this host–guest coordination pattern is no longer very stable. When the host–guest and host–phenyl contacts drop further, all components of the inter-molecular coordination become looser. From the last structure extracted on the surface, we know that most parts of the guest have left the host cavity at this position, which serves as an intermediate state along the association/dissociation pathway. As for the protonated G5, the crescent landscape depicting the conformational fluctuations is still observed under FM-PM6, but the free energy surface seems more rugged and the low-(free-)energy region is wider than the GAFF and GAFF2 cases, which suggests a higher level of conformational fluctuations and more dynamic host–guest coordination. We extracted structures from representative points to understand the detailed differences between the FM-PM6 interaction patterns and the previous results. The first structure locates at the center of the crescent minimum with intermediate values of host–guest and host–phenyl contacts. Although this position on the $C-C_{Ph}$ surface often indicates a typical center-binding pose, the flexible FM-PM6 parameter set makes the host–guest coordination different. The twisted host cavity holds the guest molecule, but the shape of the host is quite irregular. One of its -C=O portals is close to the -NH_2_^+^ group of the guest, but the angle makes it difficult to form hydrogen bonds. The second structure has larger host–guest and host-phenyl contacts than the first one. Compared with the previous center-binding pose, the host–guest coordination at this position is more compact. The host cavity is twisted to fit the guest surface in a better way, which maximizes both the host–guest and the host–phenyl contacts. The last structure has smaller contact numbers and lies at the tail of the crescent minimum. In this binding structure, the phenyl ring of the guest fluctuates outside the host cavity and becomes solvent-exposed. Overall, the basic conformational-fluctuation feature of the CB8–G5 binding observed under the GAFF derivatives remains under FM-PM6, but the host cavity is more twisted than the GAFF derivatives, which is still in agreement with the observations in the previous host–guest complexes (i.e., G1 to G4). The situation of the deprotonated form of G5 is still similar. The basic crescent free energy landscape remains. The structure extracted at the center of the crescent depicts a center-binding pose where the twisted host holds the guest in its central cavity. The other structure locates at the extra horn with larger host–guest and host-phenyl contacts. This pose is similar to that observed in the protonated G5 case, where the distorted host is in a good shape that fits the surface of the guest, thus maximizing the host–guest coordination. As for the next structurally complex guest G6, its free energy surface under FM-PM6 is very similar to that under the initial guess GAFF2. A wide free energy minimum is observed for the CB8–G6 complex, but the spread of the FM-PM6 basin is narrower than the GAFF2 one. Following the previous GAFF2 analysis, we extract two structures at the same positions as the GAFF2 case, i.e., one at the center of the wide free energy minimum and the other at the small-C tail of the basin. In the first bound structure, the 6-membered rings of the guest except the phenyl moiety are inside the host cavity, while in the second pose, the phenyl and cyclohexyl moieties fluctuate to be solvent-exposed, and only the piperidinium ring penetrates the host cavity. Therefore, it is clear that the basic picture of the host-G6 binding remains unchanged. Namely, the distorted host still holds the guest tightly and the guest is fluctuating inside the host cavity. The CB8–G7 surface under FM-PM6 is similar to the GAFF2 one, but only three bound conformations are observed under this parameter set. We can still differentiate them by their host–phenyl contacts. The first one with the largest $C_{Ph}$ features the phenyl ring inside the squashed host cavity and the other parts outside (i.e., solvent-exposed). This binding pose is also observed under the GAFF and GAFF2. However, under GAFF2 there is another pose with slightly smaller host–phenyl contacts in the neighborhood of this binding pose and the barrier between these two states is not high, but this free energy minimum is missing in the current FM-PM6 case. This missing binding pose features a more deeply embedded phenyl ring that comes out from the other side of the host cavity and becomes solvent-exposed and the other parts of the guest are closer to the host center, which leads to smaller host–phenyl contacts and larger host–guest contacts. This missing minimum has similar thermodynamic stabilities to the first pose under GAFF2 but is unstable under FM-PM6. Note that under GAFF this missing bound conformation is also of low stability, which agrees with the current FM-PM6 case to some extent. Therefore, considering this fact, the host–guest binding under FM-PM6 is somehow sharing some similarities with that produced by the GAFF parameter set. Among the three minima on the $C-C_{Ph}$ surface, the second free energy minimum has intermediate values of host–phenyl contacts but the largest host–guest contacts. The structural feature of this binding pose is that only the phenyl moiety is outside the host cavity, while all the other parts of the guest stay at the center of the host ring. Although the FM-PM6 pose is similar to the GAFF2 one, it is worth noting that the host cavity is much more twisted than the GAFF2 case, which agrees with the increased flexibility upon parameter refitting targeting PM6-D3H4X. Note that this binding pose is thermodynamically stable under GAFF2 but not under GAFF, which suggests that the host–guest interaction under FM-PM6 shares some similarities with that produced by the GAFF2 parameter set. Among the three conformational states, the last minimum has the smallest host–guest and host–phenyl contact numbers, which suggests looser host–guest and host–phenyl coordination. The basic feature of this binding pose is that the phenyl ring is far away from the host, while the other parts of the guest are relatively close to the center of the squashed host. This binding structure is formed when an intermediate level of association or dissociation is accomplished. Still, compared with the GAFF2 structure, in the FM-PM6 one the host cavity is more twisted and irregularly shaped. Note that this binding pose is thermodynamically very unstable under GAFF and of some stability under GAFF2. Considering this phenomenon, the host–guest coordination produced by FM-PM6 is again similar to GAFF2. Overall, the CB8-G7 binding under FM-PM6 shares similarities with both GAFF and GAFF2. Therefore, considering the behaviors of all host–guest pairs under investigation, the FM-PM6 parameter set produces binding behaviors similar to either GAFF or GAFF2 or both under specific circumstances. However, a worth-noting difference is that the FM-PM6 parameters tend to produce a highly flexible host cavity that is often more squashed and twisted than the other cases.

Finally, we check the $C-C_{Ph}$ free energy surfaces under the FM-BLYP parameter set in Figure 8. The reference level of this parameter set is more accurate than the previous case (i.e., PM6-D3H4X). Therefore, this FM-BLYP parameter set is considered the most accurate molecule-specific parameter set employed in the current work and is expected to produce more reliable results than the other. For the structurally simple guest G1, a wide free energy basin is observed in the typical center-binding position, where the guest fluctuates inside the widely open host cavity and the -NH_2_^+^ group forms hydrogen bonds with -C=O portals. The shape of the free energy basin or the distribution of the low-(free-)energy regions and the bound conformations extracted there are all very similar to the GAFF2 results. The situation of the host–guest interactions in the CB8–G2 case is more complex. Under FM-BLYP, the low-(free-)energy regions are significantly different from the previous GAFF, GAFF2 and FM-PM6 cases. The regions with small host–phenyl contacts that are thermodynamically favorable under GAFF2 are relatively unstable under FM-BLYP. The most stable conformational state under this parameter set has the largest $C_{Ph}$ among all low-(free-)energy regions. The structural features of the most stable binding pose include that the phenyl moiety connected to the amide bond penetrates the host cavity, while the other parts of the guest stay far away from the host center and are solvent-exposed. The wideness of this large-$C_{Ph}$ minimum suggests a significant level of conformational fluctuations, where the other solvent-exposed parts of the guest can fluctuate to bind to the surface of the host or become far from any side of the host to minimize the host–guest coordination. The other (meta-)stable binding poses all have smaller host–phenyl contacts, which suggests that the selected phenyl ring is relatively far from the host center in these binding structures. Visualization of the host–guest interactions in these regions suggests many other inter-molecular coordination patterns aside from the phenyl-in-the-cavity one. This phenomenon indicates that the FM-BLYP parameter set has a high level of tolerance of different host–guest interaction patterns and a high level of conformational fluctuations in the bound state, which makes it possible to cover almost all coordination patterns that the host–guest system could form. The CB8-G3 $C-C_{Ph}$ surface is as simple as the other cases, where a single free energy minimum featuring the whole guest inside the central cavity and the -NH^+^ and two -OH groups of the guest forming hydrogen bonds with the -C=O portals of the host is observed. Note that the FM-BLYP binding structure is similar to the GAFF and GAFF2 ones instead of the irregularly shaped twisted FM-PM6 one, which is expected considering the host behavior observed previously in unbiased simulations. The guest G4 is structurally similar to G3. The host–guest coordination under GAFF is dominated by the typical center-binding pose, while under the GAFF2 and FM-PM6 parameter sets many other bound conformations are populated non-negligibly. The refitted FM-BLYP parameter set, instead of being similar to the initial guess GAFF2 and FM-PM6 that is generated in a similar way but targets a lower-level reference, produces dominant center-binding preferences in the CB8-G4 complex. Namely, compared with the CB8–G4 binding patterns under GAFF2 and FM-PM6, the FM–BLYP situation is more reasonable and similar to the CB8–G3 case, where the inter-molecular coordination between the macrocyclic host and the structurally rigid guest is strong/tight and locates at the typical center-binding region. However, despite the structural similarities between the G3 and G4 guests, the G4 free energy surface is still a bit different from the single-minimum G3 one. Specifically, there are three free energy minima on the G4 surface. The distribution of the three minima is similar to the conformational-fluctuation crescent in the CB8–G5 case, which suggests that the FM-BLYP CB8–G4 complex also has some levels of conformational fluctuations. Visualization of bound conformations in the three minima suggests that all of them are center-binding ones, and the differences lie in the degree of distortion to the host cavity and the position of the guest inside the host cavity. The most stable binding pose locates at the middle minimum, where the guest stays at the center of the host and the host cavity is minimally perturbed. Its neighboring small-C state along the x-axis shares most of the binding features of the first minimum, and their host–phenyl contacts are similar. The difference between these two minima mainly lies in the degree of distortion to the host ring. In the small-C minimum, the host cavity is distorted, and its shape becomes relatively irregular. The other less stable minimum along the y-axis has smaller $C_{Ph}$ but larger C, which suggests that the host–guest coordination, in this case, should differ from the other poses in many aspects. The structure extracted here suggests that the host cavity is more squashed than its neighboring state and the phenyl ring of the guest is relatively far from the center of the oval host. The $C-C_{Ph}$ surface for the protonated G5 is similar to the GAFF, GAFF2 and FM-PM6 results. The basic conformational fluctuation picture remains unchanged. Namely, the whole guest stays in the central cavity, the -NH_2_^+^ group of the guest forms inter-molecular hydrogen bonds with the -C=O portals of the host, and the fluctuations of the guest inside the host cavity make some parts of it far away from the host center. As for the deprotonated form of G5, the crescent free energy landscape due to the conformational fluctuations in the bound state is still similar to the results under the other force fields. Namely, the large-C large-$C_{Ph}$ position describes a picture of the whole guest penetrating the host cavity, while the other feature parts of the guest fluctuate and leave the host center. However, compared with the previous results, the free energy basin is quite wide and there is no explicit free energy barrier along the fluctuation pathway, which suggests a high level of conformational fluctuation in the bound state. The CB8–G6 case under FM–BLYP differs significantly from the previous GAFF2 and FM-PM6. Instead of the wide free energy minimum featuring significant conformational fluctuations in the bound states observed under the previous cases, under FM–BLYP there are two narrow minima with high free energy barriers between them. This phenomenon suggests that the FM-BLYP CB8–G6 complex has two preferred bound conformations, and the interconversion between them is quite difficult. On the $C-C_{Ph}$ surface, the two narrow minima fall in the wide low-(free-)energy basin under GAFF2 and FM-PM6, which suggests only a subset of bound structures during the conformational fluctuations are stable under FM-BLYP. The bound state with larger host–guest and host–phenyl contacts is thermodynamically more favorable than the other. Due to its larger C and $C_{Ph}$, it is expected to have more compact host–guest and host-phenyl coordinations. Visualization of the structures there gives us a picture that the cyclohexyl and phenyl moieties of the guest are penetrating the host cavity, while the piperidinium ring is solvent-exposed but with its -NH^+^ group forming one inter-molecular hydrogen bond with one -C=O portal of the host. As expected, this binding pose is a snapshot during the conformational fluctuations in the bound state under the initial guess GAFF2 and the FM-PM6 parameter set that is generated in a similar way. The other thermodynamically less favorable conformational state has only the piperidinium moiety penetrating the host cavity, while the other two rings of the guest are far from the host center and solvent-exposed. We then turn to the last guest G7. Overall, the CB8–G7 $C-C_{Ph}$ surface under FM–BLYP is similar to the GAFF2 and FM-PM6 cases. However, similar to the FM-PM6 case, only a subset of the GAFF2 minima (specifically three out of four) are observed. In the minimum with the largest host–phenyl contacts, the phenyl ring is deeply inserted into the host cavity and even comes out from the other side to be solvent-exposed. The other parts of the guest are close to the host center, which leads to large host–guest contacts. Under GAFF2, conformational fluctuations around this state lead to a two-state behavior in the neighborhood of this binding conformation, while under FM-BLYP only this single fluctuation center remains. As the host behavior under FM-BLYP is very similar to the GAFF2 case, we expect this variation to arise from the changes in the bonded parameters of guest G7. In the conformational state with intermediate values of host–phenyl contacts, only phenyl leaves the host cavity, while the other parts of the guest stay at the center of the host. The structural features of this bound conformation are extremely similar to the GAFF2 ones. However, this minimum is wider under the refitted parameter set, which suggests that the degree of conformational fluctuations in this state is higher than in the GAFF2 case. In the last state on the free energy surface, the whole guest starts to depart from the host cavity. Compared with the previous binding pose, the phenyl ring becomes further away from the host cavity, and the other parts of the guest are also leaving the host center. As discussed previously, this conformation seems like an intermediate along the binding/unbinding pathway. A worth noting point for the FM-BLYP case is that, unlike the GAFF, GAFF2 and FM-PM6 cases where a binding pose is obviously more thermodynamically favorable than the other, these three binding poses are of extremely similar relative free energies (differences similar to the statistical uncertainty) under this more accurate QM-targeted system-specific parameter set. Thus, these conformational states are populated similarly or have similar weights in the conformational averaging. Overall, the FM-BLYP picture of CB8–drug coordination has its own features and differs from those produced by the other parameter sets in many aspects, but the general picture that the squashed host holds the guest tightly with a significant level of conformational fluctuations in the bound state remains unchanged. Therefore, we can safely conclude that the general picture of the CB8 host–guest binding is indeed caught in our series of works.

It should be noted that the general picture of CB8 host–guest binding summarized from our extensive computational modelling is indeed observed in many experimental structures [111,112]. For instance, a detailed structural analysis of the CB8 host–guest interaction from experimental crystal structures was reported [113]. It was observed that the host conformation in many CB8–guest complexes were indeed distorted (ellipsoidal shaped), validating our modelling results and the obtained general picture of CB8 host–guest binding.

### 2.4. Binding Thermodynamics

The dynamic behavior of the host ring and the binding-mode investigation shown above depicts the impacts of the refitting of bonded terms. The bonded interactions determine the conformational preference of each molecule. The GAFF and refitted FM-PM6 parameter sets describe a flexible CB8 ring, which can easily form a squashed conformation in the solvent itself, i.e., in the absence of the external guest. By contrast, the GAFF2 and FM-BLYP parameter sets provide a stiff host cavity that is less energetically favorable in the squashed conformation. However, in the presence of the external guest, the host cavity could still be squashed in order to form tighter host–guest coordination. Due to the difference between the energy costs of distorting the host cavity under different bonded parameter sets and the difference between the bonded terms for the external guest, the strengths of host–guest interactions are altered. First, the binding modes differ under different bonded parameter sets, which would alter the pattern of inter-molecular interactions and thus the strength of host–guest binding. Second, even the host–guest coordination pattern is unchanged, the detailed inter-molecular atom-atom distances are altered, which would also influence the strength of inter-molecular electrostatic and vdW interactions and thus the strength of host–guest coordination. Therefore, binding affinities under different bonded parameters are expected to show some differences. Below, we present a detailed comparison between the predicted binding affinities under different bonded parameter sets and the experimental reference [114]. Note that the binding free energy is estimated as the sum of the free energy difference between the global minimum and the zero-contact decoupled state from the radius-contact surface and the entropic correction that is used to recover the standard-state definition, as discussed in the computational details Section 2.2 and our previous works [78,79].

The predicted binding thermodynamics and the corresponding statistical uncertainties (standard deviation, SD) under the GAFF derivatives (i.e., GAFF and GAFF2) and the refitted FM-PM6 and FM-BLYP parameter sets are summarized in Table 1. The GAFF results are extracted from the best RESP predictions in our previous work, where iterative refitting of the atomic charges is performed to calibrate the charge quality [78]. Note that these atomic charges are also used in the current newly generated GAFF2, FM-PM6 and FM-BLYP datasets. Thus, these predictions only differ in the parameters of the bonded terms. We use three error metrics including the mean signed error (MSE), the MAE, and the RMSE to evaluate the deviations of the predicted binding affinities from the experimental reference, while for the consistency of the predicted and the experimental ranks of binding thermodynamics Kendall’s rank correlation coefficient (τ) and Pearlman’s predictive index (PI) are employed. Similar to the previous work, the final estimate of CB8–Ketamine affinity is the average of the protonated and deprotonated results, due to the closeness of the experimental pKa (pH 7.5) and the pH condition that the binding affinities are estimated (pH 7.4).

We first check the results obtained with the existing general force field parameters for drug-like molecules, i.e., the GAFF derivatives. Comparing the GAFF and GAFF2 estimates, it is clear that these two GAFF force fields provide different host–guest affinities. For G1, G2, protonated and deprotonated G5, and G6, the GAFF2 estimates are higher and also closer to the experimental data than the GAFF results. By contrast, for G3, G4 and G7, the use of GAFF2 parameters leads to smaller binding affinities than the GAFF estimates. For the specific guests G4 and G7, the GAFF2 estimates are significantly worse than the GAFF ones, which is the main source of error in the whole dataset. As a result, although GAFF2 improves the results for most guests, this significant problem results in a bit higher mean errors (e.g., RMSE), which suggests that the GAFF2 is actually of a bit lower performance compared with GAFF. Interestingly, the ranking coefficients are almost unchanged. The correlation between the modelling results and the experimental values is shown in Figure 9. We can see that although the GAFF2 estimates are closer to the y = x line for five systems compared with GAFF, there are two guests that are obviously farther from the diagonal than GAFF. There are only two points locating outside the ±2 kcal/mol line of the experimental results under GAFF, while the number becomes 3 under GAFF2. Namely, there are more ‘bad’ points under the GAFF2 parameter set. This behavior could, to some extent, help in understanding the increase in RMSE when transferring from GAFF to GAFF2. Overall, considering the small differences between the error and ranking metrics for the GAFF derivatives, their performances on the prediction of binding affinity are similar for the CB8 host–guest systems.

We then turn to the refitted GAFF2 parameter sets. For the lower target level PM6–D3H4X, compared with the initial guess GAFF2, the predicted binding affinities are improved for G1, G2, and protonated G5, remain almost unchanged for G3, and become worse for G4, deprotonated G5, G6 and G7. Overall, the error metrics obtained under the FM-PM6 parameter set are similar to the GAFF2 results. However, obvious differences are observed for the ranking coefficients. The Kendall τ of FM-PM6 is significantly smaller than GAFF2, while the PI is only marginally lowered. The prediction-experiment correlation for this dataset is also presented in Figure 9. We can see that for most systems, the GAFF2 and FM-PM6 predictions are similar. The number of ‘bad’ points with deviations larger than ±2 kcal/mol from the experimental reference is three under this parameter set, and these three systems are exactly the outliers in the GAFF2 case. Therefore, the SQM-targeted refitting procedure cannot correct the significant problems of the initial guess GAFF2 in modelling these guests. For the whole dataset, the betterment and decline are of similar magnitudes and the net change is minimal. The above observations suggest that although the refitted force field reproduces the energetics at this computationally efficient semi-empirical level in a better way, it may fail to improve the quality of predictions and produce higher prediction-experiment correlations.

When a higher level BLYP-D4/def2-SVP is selected, we reach the FM–BLYP dataset. This target level is more accurate than the previous PM6–D3H4X, and thus this refitted FM-BLYP parameter set should be more accurate than the previous FM-PM6. As the FM-BLYP parameters are generated in a system-specific way, they are also considered to be better than the pre-fitted transferable GAFF2 parameters. Checking the statistics in Table 1, we can see that the predicted binding affinities under this parameter set are indeed significantly improved compared with the initial guess GAFF2 and the low-level FM-PM6. Compared with the original GAFF2 results, the predicted binding free energies are improved for all guests except G6. For this specific guest, the original GAFF2 prediction is in perfect agreement with the experimental value, but the deviation of the FM-BLYP result is rather significant. The FM-BLYP results are also significantly better than the FM-PM6 ones, which is expected according to the higher accuracy of the target level. As for the error metrics, the RMSE under this parameter set reaches a very low value of 1.8 kcal/mol, which is much smaller than all other predictions reported so far. The MSE is close to zero, which suggests that there is no systematic overestimation or underestimation observed in the FM-BLYP predictions. The MAE approaches 1.1 kcal/mol, which is also the lowest value observed so far. Significant improvements are also observed for the two ranking coefficients. The correlation of the computed and experimental values under the FM-BLYP parameter set is provided in Figure 9. The number of ‘bad’ points falling outside the ±2 kcal/mol line from the experimental reference is still two. However, for the other five guests, the agreements between the modelling and experimental results are almost perfect. Overall, compared with the other modelling schemes, the combination of RESP charges and FM-BLYP parameters achieves a high-level accuracy, and the predictive power of computational tools in challenging host–guest systems is already in sight.

### 2.5. Guidelines for Host–Guest Modelling

Accurate descriptions of host–guest interactions require accurate inter- and intra-molecular potentials. The inter-molecular interactions are determined by non-bonded electrostatics and vdW terms, with the former (i.e., electrostatics) being more significant. The intra-molecular interactions involve bond stretching, angle bending, and dihedral terms, and the dihedral term plays the most crucial role in determining the conformational preference of each molecule. The interplay between the non-bonded and bonded interactions is the core of molecular recognition. A balanced description of inter- and intra-molecular interactions is needed.

In our previous work focusing on the non-bonded terms, the charge schemes widely used in drug discovery are assessed in great detail and the best-performing regime is identified [78]. The RESP charge scheme could reproduce the Coulombic molecular ESP very well, leading to accurate descriptions of inter-molecular host–guest interactions. However, the conformational dependence of the RESP charge scheme is quite significant, while the AM1-BCC charge scheme is more mean-field-like and provides similar results for different charge-generation configurations. Considering this fact, the binding thermodynamics obtained with RESP charges could be inaccurate or wrong when the charge-generation configuration is improperly selected. A solution to this problem recommended in our previous work is using an iterative procedure to parameterize the electrostatics. Specifically, the mean-field-like AM1-BCC model as an initial guess could be used to explore the conformational space, after which the most stable bound conformation is extracted to re-parameterize the molecule with the RESP scheme. Note that the RESP charge scheme could also be used in the initial exploration of the configurational space. Although the RESP scheme is accurate for most systems, it still suffers from the lone-pair problem faced by all fixed-charge models. To further improve the description of electrostatics, polarizable models or even QM calculations should be considered [49,86,115,116,117,118,119,120,121,122].

In this work, we consider another important influencing factor in MM potentials, the bonded terms. The bonded interactions are not directly involved in the calculation of inter-molecular non-bonded interactions, but still play a crucial role by influencing the intra-molecular conformational preference of each molecule involved in inter-molecular recognitions. The transferable GAFF derivatives for drug-like molecules are compared and assessed, and system-specific parameters are obtained by refitting the bonded terms (bond stretching, angle bending, dihedral flipping) with the generalized FM scheme, which considers both the atomic force, the energy, and various regularization terms to obtain a reasonable set of parameters that reproduce the properties at the target level. The target levels determine the ultimate accuracy that the refitted model can achieve. Here, we selected the dispersion, hydrogen-bonding, and halogen-bonding corrected semi-empirical level PM6-D3H4X and the frequently used and dispersion corrected ab initio level BLYP-D4. Compared with the original GAFF2, the energetics and atomic forces at the target levels are reproduced in a much better way under the refitted parameter sets. Further improvements could be expected by choosing a higher-level QM target. For instance, with more computational resources, we could use a larger basis set (e.g., def2-TZVP) and/or more accurate functionals such as composite methods and double hybrids. Further, although the refitted model outperforms the original GAFF2 parameter set in reproducing the energetics at the target QM level, obvious deviations could still be observed. On this aspect, more complicated energy functions (formula) could be employed to improve the fitting quality.

With the constructed models to describe the energetics, the solvated host–guest systems could be simulated. Due to the high (free) energy cost of breaking the host–guest coordination and the complex conformational ensemble to explore, some enhanced sampling techniques need to be employed in the modelling of host–guest binding. As for the selection of the configurational space to explore, using some CVs to describe the relative position of the host and the guest seems useful in order to scan their binding patterns. Based on our series of works on host–guest modelling [78,79,92], the spherical-coordinates-biased protocol seems sufficient for converged sampling within μs-length simulations, although some more CVs should be biased or coupling the simulation with other sampling techniques should be considered if hidden barriers on orthogonal CVs (e.g., internal motions) hinder the convergence of the current protocol.

Overall, for model construction, the combination of the RESP charges and the refitted molecule-specific BLYP-D4 bonded parameters shows the best performance in CB8 host–guest binding. For sampling the configurational space, the spherical-coordinates-biased protocol is sufficient. Thus, we recommend using a similar protocol in the modelling of CB8-related molecules and more generally similar host–guest systems.

### 2.6. Further Improvements

According to our series of works on CB8 host–guest binding simulations, the combination of the RESP charges and some refitted bonded terms provides a balanced accuracy-cost regime for host–guest modelling. The fixed-charge model augmented by an optional iterative refitting procedure provides accurate electrostatics and intra-molecular conformational preferences. If the force-field refitting takes too much computational resources, the GAFF derivatives are also usable. The calculated binding thermodynamics and interaction patterns with the combined model achieve a high level of accuracy within the framework of molecular simulations with classical force fields. The sampling issue is mostly solved by enhanced sampling techniques coupled with a 3D spherical CV set. Although the sampling and Hamiltonian issues are solved mostly, there are still some places where further improvements could be performed. Below, we provide some discussions about several aspects that further developments could be performed.

We first consider the sampling of the configurational space. If there are some hidden barriers on orthogonal degrees of freedom, more CVs (e.g., some selected dihedral) could be added to the CV set or coupling other enhanced sampling schemes (e.g., replica exchange) with the current scheme could be considered. Note that this is not really a problem for the current CB8 host–guest system, which has been discussed and tested in our previous work [78]. Aside from altering the enhanced sampling regime, modifications on other simulation details (e.g., basic MD parameters) could also be considered. For instance, for the integration of the equations of motion, the widely employed splitting arrangement (i.e., the leapfrog integrator) could be replaced with more efficient schemes such as BAOAB [31,32,123] to achieve higher accuracy in the configurational space. On this aspect, further improvements could be achieved through multiple time scale/step algorithms [33,34,124,125].

We then consider the description of the system or Hamiltonian, which is actually more problematic than the sampling issue. The Hamiltonian used in the current work is the classical force field, which can be improved by using more accurate forms of potential functions. The electrostatics could be improved by introducing multipoles or fluctuating charges, which provides a better description of the polarization effects that are often claimed to be significant [126,127]. As this polarization-related issue has been discussed in our previous work, we focus on the forms of bonded terms in the current work. The AMBER derivatives describe the bond stretching and angle bending with harmonic potentials, which could be improved by using more complex functional forms (e.g., cubic or quartic terms) and introducing cross terms to account for the coupling between different motions [128,129,130,131,132]. An example of this direction is the Merck molecular force field (MMFF) series. Specifically, MM2 has bond-angle (stretch-bend) cross terms [133], MM3 further adds stretch-torsion and bend-bend interactions [134], and the MM4 functional form further includes stretch-stretch, torsion-bend, bend-torsion-bend, torsion-torsion, torsion-improper torsion, and improper torsion-torsion-improper torsion interactions [135]. A recent review summarizes that the MMFF series produce better results than other classical force fields in a number of applications [136]. Another example of force fields with more complex functional forms is the consistent force field (CFF) [137,138,139], which is often considered to be the origin of modern force fields. The consistency in its name suggests its aim of transferability. Namely, the functional form and parameters of this force field are designed to simultaneously account for a large number of compound types. CFF incorporates an extensive set of coupling terms including stretch-stretch, bend-bend, stretch-bend, bend-bend-torsion and out-of-planes coupling terms [137,138,139]. It should be noted that the modification of the force-field expression is not limited to the host and the guest molecules. The other components included in the simulation box, i.e., ions and water molecules, also influence the host–guest coordination and can also be improved. For instance, replacing the simple TIP3P water with more accurate models (e.g., OPC [140]) could improve the accuracy of the solvent model and hopefully also the solute-solvent interaction. However, we should keep in mind that the computational cost is also increased significantly upon the force-field upgrade. Another direction to further improve the Hamiltonian is introducing nuclear quantum effects (NQEs) via the path-integral formulation [141,142,143,144,145,146]. The influence of the NQEs on the binding thermodynamics is mostly on hydrogen bonds, the interaction strength of which could alter by about 0.5 kcal/mol. Note that hydrogen bonds and thus NQEs are quite general in all host–guest interactions, not just the current CB8 host–guest binding.

## 3. Methodology and Computational Details

### 3.1. Model Construction

The 3D structure of the host CB8 and the 2D chemical structures of the guest molecules are shown in Figure 1a. The experimental binding affinities [114] of these host–guest systems are summarized in Table 2. In model construction, we first need to determine the protonation states of each molecule. The experimental and ChemAxon-predicted pKa values of each molecule are summarized in Table 2, with which the protonation state of each molecule is determined. The host CB8 is net-neutral, and all guest molecules are protonated except G5 Ketamine. Due to the closeness of the experimental and ChemAxon predicted pKa of this molecule and the pH condition (pH 7.4) that the experimental binding affinities are measured, there should be an acid-base equilibrium in the CB8–G5 binding. Thus, both the protonated and deprotonated forms of G5 are modelled in our computational investigation. The net charges of the host and the guest molecules are summarized in Table 2.

According to our previous work on CB8–drug systems, the reproduction of the molecular ESP is better with the restrained electrostatic potential (RESP) [147] charge scheme than with the corrected semi-empirical AM1-BCC [148]. Thus, the inter-molecular electrostatic interactions are more accurate with RESP charges. Further, the deviations of the predicted binding thermodynamics from the experimental values are smaller with RESP charges than with AM1-BCC and the ranking coefficients are also better for the RESP charge scheme. Therefore, we employ the RESP charge scheme to represent electrostatics in the current work. The performance of the RESP charge scheme shows obvious dependence on the guest conformation used in charge generation [78], and thus the iteratively calibrated charge set [78] is chosen. The other force-field parameters including the vdW interactions and the bonded terms are obtained from GAFF (GAFF version 1.81) or the second generation of GAFF (GAFF version 2.11, GAFF2) [88]. As the GAFF derivatives are parameterized to achieve high transferability, their accuracies for specific systems could be relatively low compared with parameters fitted for specific systems. To further improve the quality of the model, we also parameterize system-specific force fields for the bonded terms to achieve a higher level of accuracy. The generalized FM scheme [93,94] is used in this bonded-term refitting, the details of which would be discussed in the results part. The starting structure of each host–guest complex is obtained by simply superposing the center of masses (COM) of the host and that of the guest. Solvation is performed with TIP3P [149,150] water and the minimum distance between the solute surface and the box edge is set to 25 Å, which is sufficiently large to define a decoupled state. The truncated octahedron cell is replicated in the whole space with periodic boundary conditions. Non-polarizable monovalent spherical counter ions [151,152] of Na^+^ or Cl^−^ parameterized for TIP3P water are added for neutralization.

### 3.2. Free Energy Simulation

As the macrocyclic host CB8 is structurally symmetric and much simpler than biomacromolecular receptors (e.g., protein), it is often presumptive that there is only one stable binding mode in each CB8–guest complex [21,70,71,72,73,74]. However, the validity of such a priori assumption for practical systems remains unknown unless extensive exploration of the configurational space is performed. As observed in our previous extensive modelling of host–guest systems [78,79,92,153], the CB8–guest complex has a quite complex conformational ensemble, where multiple binding modes with similar thermodynamic stabilities would contribute significantly. Despite the symmetric structural feature of the macrocyclic host, the conformational fluctuations of the host and the guest, the orientation of specific functional groups, and the diverse structural features of the guest molecules would make converged sampling very difficult. The high dimensionality of the host–guest system and the high free energy barrier to overcome when exploring the decoupled state call for the use of enhanced sampling techniques. To improve the sampling efficiency, we employ well-tempered metadynamics [154,155,156,157]. Then, another question is how to deposit or distribute the biasing potential in order to explore the space of host–guest binding modes extensively. To avoid any priori of the relative host–guest position and thus the detailed interaction feature of binding modes, the biasing CVs are chosen as the spherical coordinates (ρ,θ,φ) defined by the COMs of the host and the guest [78]. When calculating the absolute binding affinity, an entropic correction term is also needed to recover the standard-state condition. For more detailed discussions about the sampling protocol, please refer to our previous works on similar host–guest systems [78,79,92]. An illustration of the 3D spherical CV is presented in Figure 1b.

To provide a detailed analysis of the host–guest coordination pattern, we calculate the contact number between groups of atoms, which is defined as $C=\sum_{i\in {\mathrm{group}}_{A}}\sum_{j\in {\mathrm{group}}_{B}}\frac{1-{\left(\frac{r_{ij}}{r_{0}}\right)}^{6}}{1-{\left(\frac{r_{ij}}{r_{0}}\right)}^{12}}$. Here, group A and group B are two groups of atoms involved in inter-molecular coordination. The distance constant in the switching function $r_{0}$ is set to 6 Å, which is consistent with our previous CB8 host–guest work [78]. The variable $r_{ij}$ represents the distance between the *i*th atom in group A and the *j*th atom in group B. To monitor the closeness of the host and the guest during the exploration of the configurational space, we calculate the total contact number between the host and the guest $C_{\mathrm{host}\text{–}\mathrm{guest}}=\sum_{i\in \mathrm{host}}\sum_{j\in \mathrm{guest}}\frac{1-{\left(\frac{r_{ij}}{r_{0}}\right)}^{6}}{1-{\left(\frac{r_{ij}}{r_{0}}\right)}^{12}}$. More detailed information on host–guest coordination (e.g., atoms involved in the inter-molecular coordination) could be extracted from its by-host-atom decomposition $C_{i}=\sum_{j\in \mathrm{guest}}\frac{1-{\left(\frac{r_{ij}}{r_{0}}\right)}^{6}}{1-{\left(\frac{r_{ij}}{r_{0}}\right)}^{12}}$, i.e., the atom–guest contact for the *i*th atom of the host. Due to the complexity of the host–guest interactions, the total host–guest contact is observed to be insufficient to differentiate different inter-molecular interaction patterns properly [78]. Therefore, according to the chemical intuition that the large cavity of the CB8 ring needs to be filled with some parts of the guest and the fact that there is at least one 6-membered aromatic ring in each guest, we consider the contact number between the host and the aromatic ring $C_{\mathrm{host}\text{–}\mathrm{phenyl}}=\sum_{i\in \mathrm{host}}\sum_{j\in \mathrm{phenyl}}\frac{1-{\left(\frac{r_{ij}}{r_{0}}\right)}^{6}}{1-{\left(\frac{r_{ij}}{r_{0}}\right)}^{12}}$ to analyze detailed features of the binding modes. As there are two phenyl rings in G2, the selected phenyl ring is directly connected to the amide bond. Note that in free energy analysis, only heavy atoms are included in the calculation of contact numbers.

As has been discussed in the model construction part, the starting configuration of each host–guest complex is obtained by simply superposing the COMs of the host and the guest. Although such a bound conformation could be energetically unfavorable and the system could be far from equilibrium, we perform 5000 steps minimization, 400 ps NVT equilibration, and 2 ns NPT equilibration to let various degrees of freedom relax and find their favorable regions. After that, we initiate 1000 ns spherical-coordinates-biased metadynamics simulations. The parameters for the metadynamics setting remain the same as in our previous work [78,79,92]. Namely, we use a 0.24 kcal/mol initial height of the repulsive Gaussian, a deposition interval of 0.5 ps, a bias factor of 20, and Gaussian widths of 0.1 nm, $\frac{\pi }{16}$, and $\frac{\pi }{8}$ for the three polar coordinates, respectively. The simulation is performed with GROMACS 2019.6 [158] patched with PLUMED 2.7.0 [159]. The velocity rescaling algorithm [160] is employed for temperature regulation at 298 K and the Parrinello–Rahman barostat [161,162] is used for pressure regulation. The leapfrog integrator with a 1 fs time step is used to propagate dynamics. Long-range electrostatics are treated with the smooth particle-mesh Ewald [163] method.

## Figures and Tables

**Figure 1 molecules-28-03124-f001:**
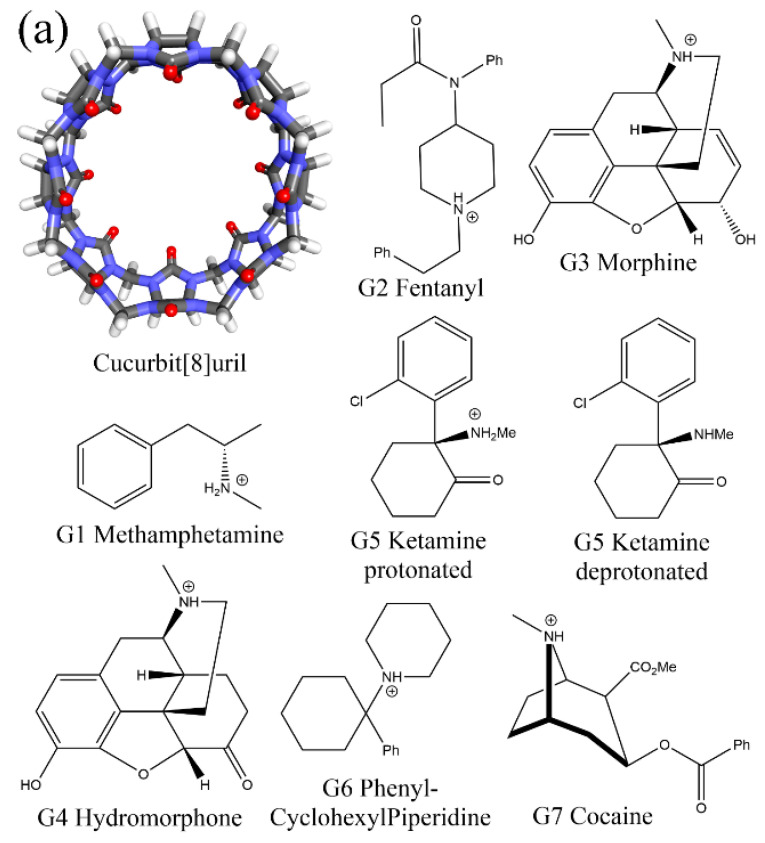
(**a**) The 3D structure of the CB8 host and the 2D chemical structures of its 7 guests. For the guest G5 Ketamine, due to the small difference between the experimental and ChemAxon predicted pKa and the pH condition that the binding affinity is measured, both the protonated and deprotonated forms of the guest G5 are considered in our modelling. (**b**) An illustration of the 3D spherical coordinates CV used to bias the simulation.

**Figure 2 molecules-28-03124-f002:**
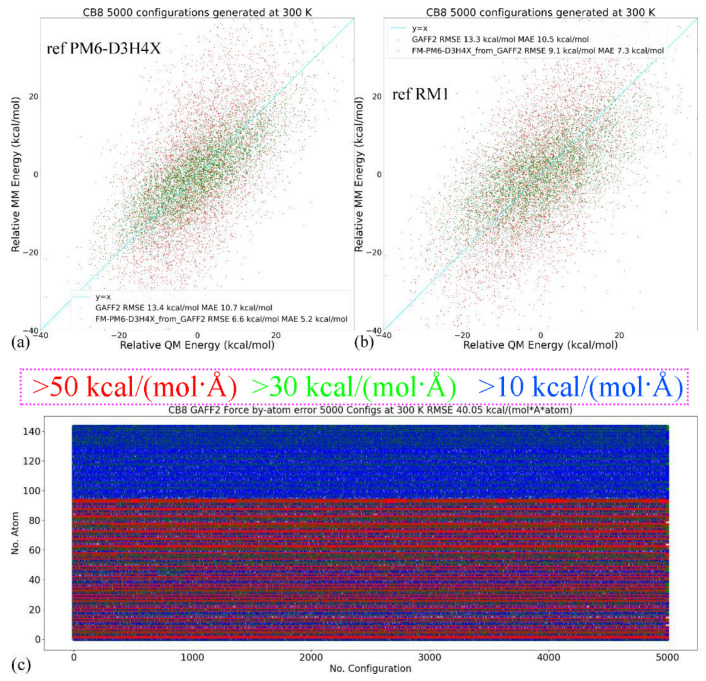
(**a**) The correlations between the MM (i.e., GAFF2 and FM-PM6) and PM6-D3H4X energetics calculated from 25 ns trajectories generated at 300 K in vacuo for the host, (**b**) those between energetics produced by MM force fields and a different SQM Hamiltonian of RM1 instead of the fitting target PM6-D3H4X. (**c**,**d**) The time series of force errors (i.e., Frobenius norm of ΔF) produced by GAFF2 and the refitted FM-PM6 force field calculated with configurations generated at 300 K for the host CB8. Red dots for the force errors larger than 50 kcal/(mol·Å), green for force errors larger than 30 kcal/(mol·Å), blue for errors larger than 10 kcal/(mol·Å), and white for the other small-error points. The first 96 atoms are non-hydrogen heavy atoms, the force errors of which are larger than the other hydrogen atoms. The errors of (**e**) the energetics and (**f**) atomic forces produced by GAFF2 and the refitted FM-PM6 force field calculated with configurations generated at 300 K for all host and guest molecules under investigation.

**Figure 3 molecules-28-03124-f003:**
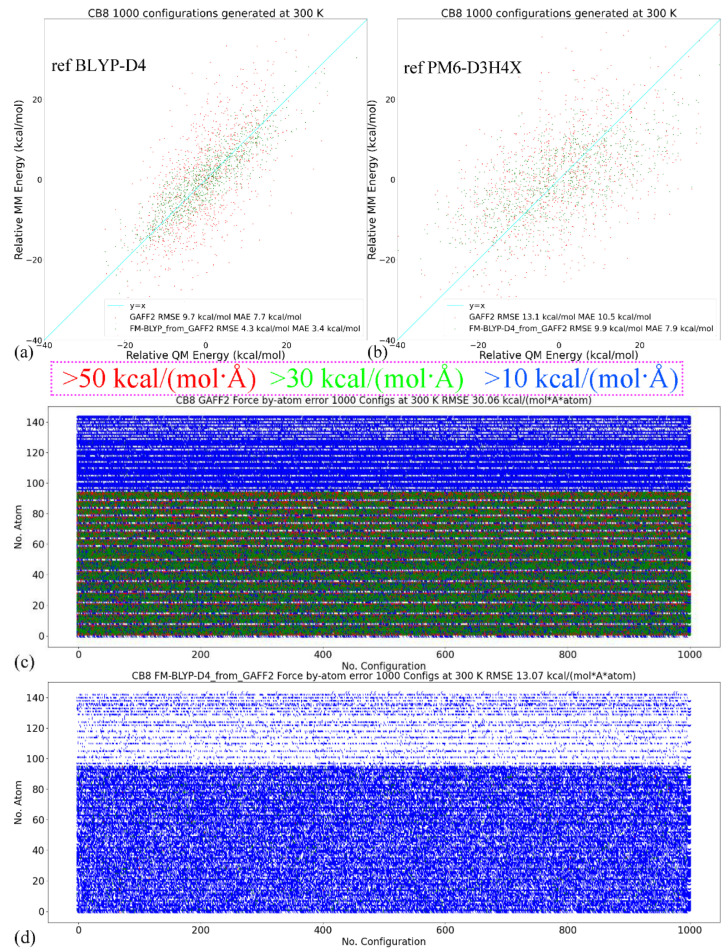
(**a**) The correlations between the MM (i.e., GAFF2 and FM-BLYP) and BLYP-D4 energetics calculated from 10 ns trajectories generated at 300 K in vacuo for the host, (**b**) those between the energetics produced by MM and a different reference level of PM6-D3H4X (i.e., the previous target in FM-PM6 refitting) instead of the target level BLYP-D4. Note that for this reference level, the FM-BLYP parameter set produces larger errors compared with the previous FM-PM6, which is fitted directly to the data at this reference level. (**c**,**d**) The time series of errors of atomic forces produced by GAFF2 and the refitted FM-BLYP force field calculated with configurations generated at 300 K for the host CB8. Red dots for the force errors larger than 50 kcal/(mol·Å), green for force errors larger than 30 kcal/(mol·Å), blue for errors larger than 10 kcal/(mol·Å), and white for the other small-error points. The first 96 atoms are non-hydrogen heavy atoms, the force errors of which are larger than the other hydrogen atoms. The errors of (**e**) energetics and (**f**) atomic forces produced by GAFF2 and the refitted FM-BLYP force field calculated with configurations generated at 300 K for all host and guest molecules under investigation.

**Figure 4 molecules-28-03124-f004:**
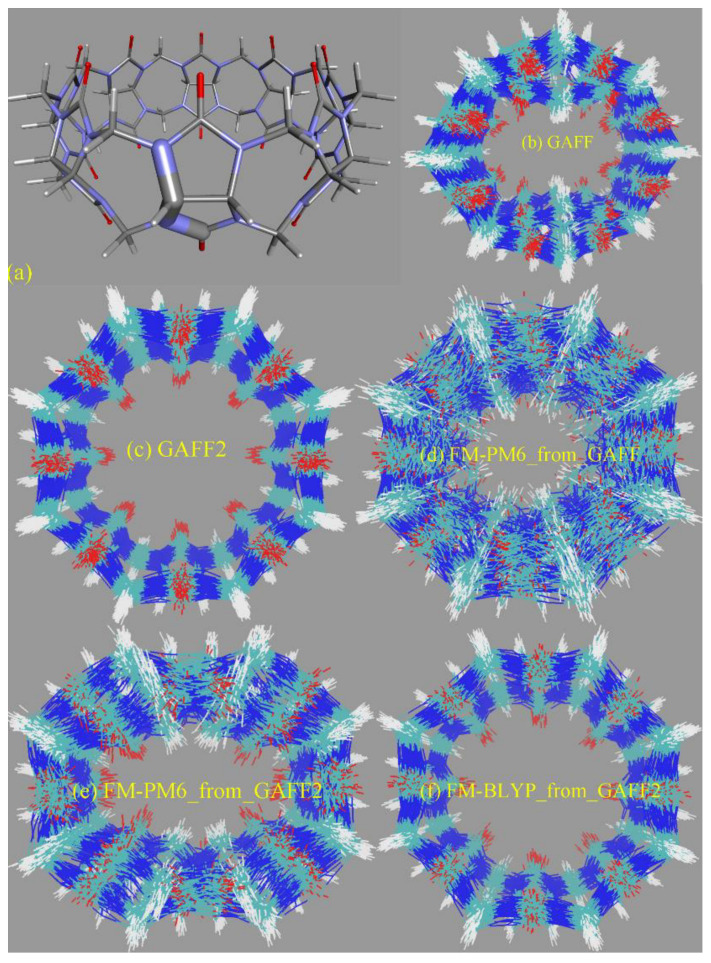
(**a**) The dihedral term that shows the most significant difference between GAFF derivatives describing the conformational preference (stiffness) of the host. The superposition of the host configurations during 20 ns unbiased simulations in explicit solvent with bonded parameters from (**b**) GAFF, (**c**) GAFF2, (**d**) the FM-PM6 parameters initiated from GAFF, (**e**) the FM-PM6 parameters initiated from GAFF2, and (**f**) FM-BLYP initiated from GAFF2. (**g**) The time series of the radius of gyration of the CB8 ring under the GAFF2 and the refitted parameter sets.

**Figure 5 molecules-28-03124-f005:**
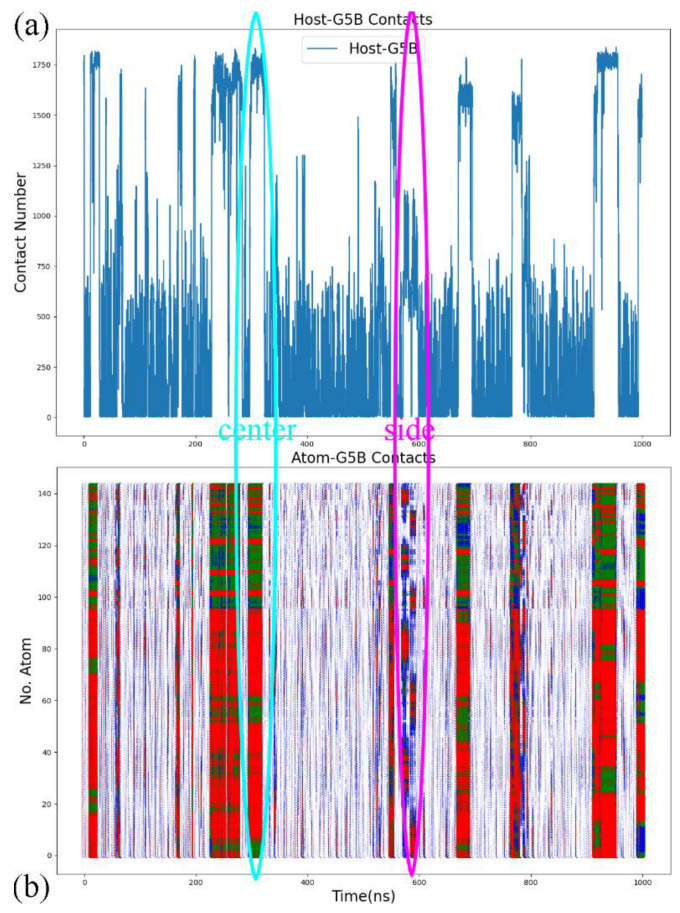
(**a**) The number of contacts between all atoms of the host CB8 and the guest deprotonated G5 and (**b**) its decomposition by each atom of the host during 1000 ns enhanced sampling simulations with the newly refitted FM-PM6 parameters. The y-axis represents the serial number of host atoms. The first 96 atoms of the host are heavy atoms, and the others are hydrogen atoms. Red dots denote contacts larger than 10, green dots represent contact numbers between 5 and 10, blue ones are those larger than 1, and the other are represented by white dots. The cyan oval provides an example of the typical center-binding pose explored during enhanced sampling simulations, while in the magenta one there is a side-binding mode.

**Figure 6 molecules-28-03124-f006:**
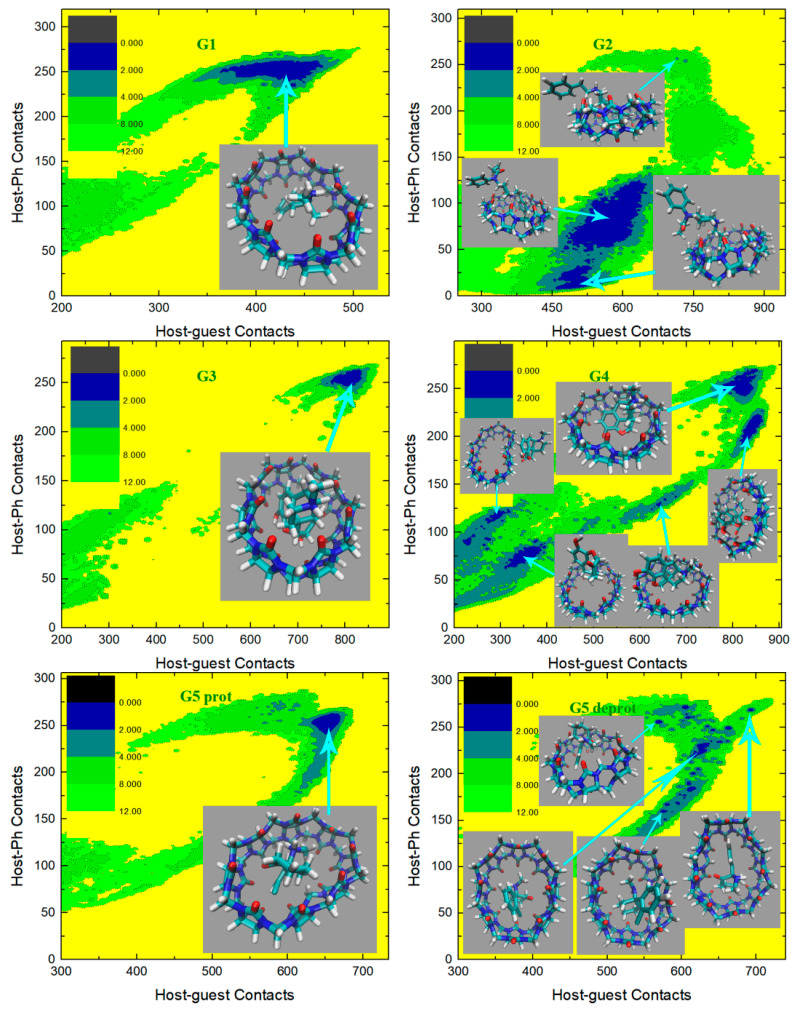
2D $C-C_{Ph}$ free energy surfaces in kcal/mol obtained under the GAFF2 force field.

**Figure 7 molecules-28-03124-f007:**
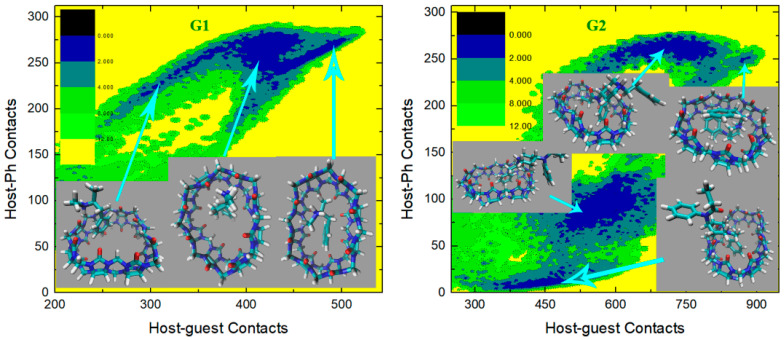
2D $C-C_{Ph}$ free energy surfaces in kcal/mol obtained under the newly refitted force field FM-PM6.

**Figure 8 molecules-28-03124-f008:**
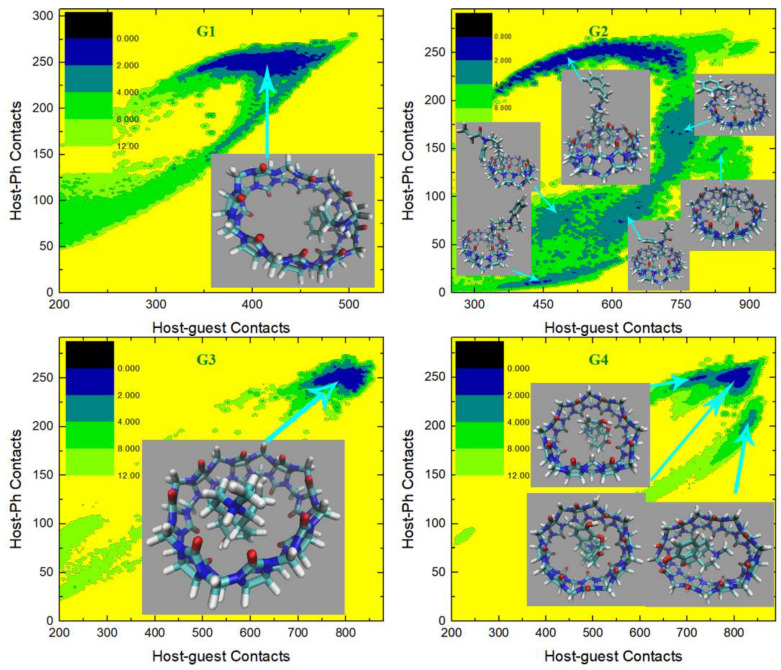
2D $C-C_{Ph}$ free energy surfaces in kcal/mol obtained under the newly refitted force field FM-BLYP.

**Figure 9 molecules-28-03124-f009:**
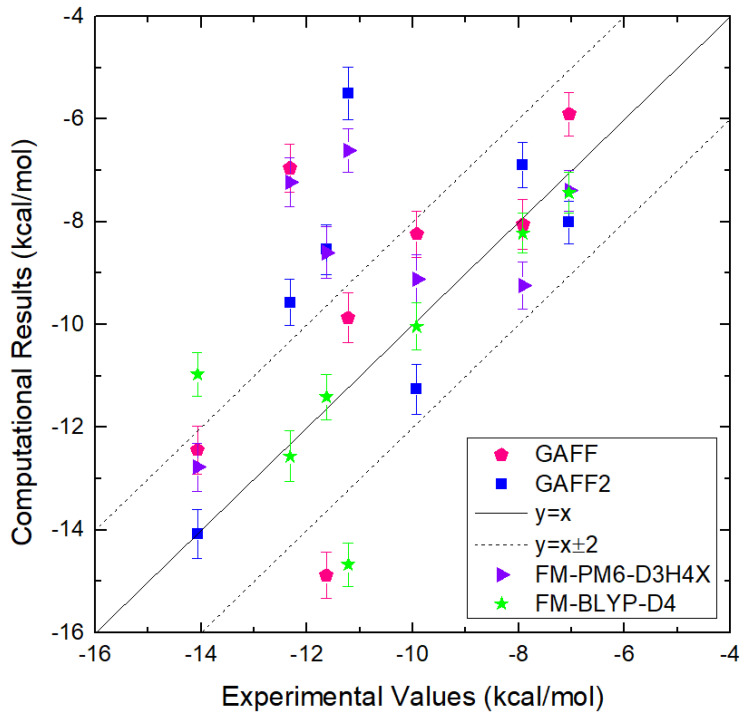
Correlation between the binding affinities obtained from our computational modeling and experimental reference for CB8–guest systems. The exact values of the binding affinities are presented in Table 1.

**Table 1 molecules-28-03124-t001:** The CB8–guest binding affinities in kcal/mol. $\Delta G_{\exp}$ is the experimental value. $\Delta G_{\mathrm{GAFF}}$ and $\Delta G_{\mathrm{GAFF}2}$ denote the binding affinities obtained with the existing general force fields of GAFF and GAFF2, respectively, while $\Delta G_{\mathrm{FM}\text{-}\mathrm{PM}6}$ and $\Delta G_{\mathrm{FM}\text{-}\mathrm{BLYP}}$ represent the binding free energies obtained with our newly fitted bonded parameter sets targeting PM6-D3H4X and BLYP-D4/def2-SVP, respectively. SD represents the standard error of the free energy estimate, which is obtained from block analysis. MSE, MAE, RMSE, τ, and PI serve as quality measurements. Due to the closeness of the pKa of G5 Ketamine and the pH condition that the experimental binding affinities are measured, the protonated and deprotonated forms of G5 are considered to be approximately equally populated. As a result, the final estimate of CB8-G5 binding affinity is the average of the protonated and deprotonated G5 results, which is used to calculate the quality metrics, while protonation-dependent results are presented for a detailed comparison of the simulated results.

Host	Guest	Δ*G*_exp_	Δ*G*_GAFF_	SD	Δ*G*_GAFF2_	SD	Δ*G*_FM-PM6_	SD	Δ*G*_FM-BLYP_	SD
CB8	G1	−7.05	−5.9	0.4	−8.0	0.4	−7.4	0.4	−7.4	0.4
	G2	−9.93	−8.2	0.5	−11.3	0.5	−9.1	0.5	−10.0	0.5
	G3	−11.63	−14.9	0.4	−8.5	0.5	−8.6	0.5	−11.4	0.4
	G4	−11.22	−9.9	0.5	−5.5	0.5	−6.6	0.4	-14.7	0.4
	G5 prot	−12.32	−3.5	0.5	−8.3	0.5	−6.4	0.5	−9.2	0.5
	G5 deprot	−12.32	−10.4	0.5	−10.9	0.5	−8.0	0.5	−16.0	0.5
	G6	−14.07	−12.4	0.5	−14.1	0.5	−12.8	0.5	−11.0	0.4
	G7	−7.92	−8.1	0.5	−6.9	0.4	−9.2	0.5	−8.2	0.4
RMSE			2.6		2.8		2.9		1.8	
MSE			−1.1		−1.5		−1.9		0.2	
MAE			2.1		2.1		2.4		1.1	
τ			0.4		0.4		0.0		0.5	
PI			0.6		0.6		0.5		0.7	

**Table 2 molecules-28-03124-t002:** The names of the CB8 host and 7 drugs/guests, the experimental binding affinities in kcal/mol, the pKa values of all guests determined experimentally at 298 K and via ChemAxon, and the net charges considered in the current work. The net charge of each molecule is determined by the relative magnitude of its pKa and the experimental condition (pH 7.4) that the binding thermodynamics is measured. The experimental and ChemAxon predicted values of all guests except G5 (Ketamine) are obviously larger than the experimental condition pH 7.4. Thus, all of them are protonated at the nitrogen atom in our modelling. As for guest G5, both the experimental and the ChemAxon results indicate the existence of acid-base equilibria. Thus, both the protonated and deprotonated forms are considered.

Molecule	Name	$\Delta G_{\exp}$	$pK_{a,\exp}$	$pK_{a,\mathrm{ChemAxon}}$	Net Charge Considered
CB8	Cucurbit[8]uril	-	-	-	0
G1	Methamphetamine	−7.05	9.87	10.21	1
G2	Fentanyl	−9.93	8.99	8.77	1
G3	Morphine	−11.63	8.21	9.12	1
G4	Hydromorphone	−11.22	8.2	8.59	1
G5	Ketamine	−12.32	7.5	7.45	0 or 1
G6	PhenylCyclohexylPiperidine	−14.07	8.29	10.56	1
G7	Cocaine	−7.92	8.61	8.85	1

## Data Availability

The data that support the findings of this study are available from the corresponding author upon reasonable request.

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
