# Peer review of "A General Picture of Cucurbit[8]uril Host–Guest Binding: Recalibrating Bonded Interactions"

_molecules, 2023, doi:10.3390/molecules28073124_

Round 1
Reviewer 1 Report
In this manuscript, the authors studied the MM force field affects on studying of host-guest interactions. By re-parametrizing of bonded parameters in GAFF using different QM level of theory, they obtained different level of performances on reproducing experimental binding affinities. Some comments are listed below for authors to consider.
1. Can the authors further explain which level of QM theory was used for fitting of the original GAFF force fields and compare that with PM6 used in the current work? The PM6 level results are so different from others considering the geometry of CB8 and PM6 refitted GAFF does not introduce significant benefit on reproducing the experimental binding profiles.
2. The authors used a contact number metric to describe the spatial relationshio between the host and guest and also mentioned that it is insufficient to differentiate different inter-molecular interaction patterns properly, thus they also used another contact number focusing on host-phenyl contacts. Did the authors try other simple metrics such as center-of-mass distance to complement the analysis? Distance between COMs can define how far away the two molecules are seperated while contact number tells the possible interaction density.For similar contact number, COM can help differentiate side interactions from in-cavity interactions.
3. Pay attention to typos, e.g. “complexity of the host-host interactions”should be “complexity of the host-guest interactions”.
Author Response
Submission ID: molecules-2241132
Responses to Referees’ comments
------------------------------------
COMMENTS TO AUTHOR:
Reviewer #1:
In this manuscript, the authors studied the MM force field affects on studying of host-guest interactions. By re-parametrizing of bonded parameters in GAFF using different QM level of theory, they obtained different level of performances on reproducing experimental binding affinities. Some comments are listed below for authors to consider.
- Can the authors further explain which level of QM theory was used for fitting of the original GAFF force fields and compare that with PM6 used in the current work? The PM6 level results are so different from others considering the geometry of CB8 and PM6 refitted GAFF does not introduce significant benefit on reproducing the experimental binding profiles.
Response: Selection of the QM level is a critical issue and the reviewer’s comment is fully reasonable. For the QM level for GAFF parametrization, the original reference (Vol. 25, No. 9 • Journal of Computational Chemistry) suggests that GAFF uses MP2/6-31G*. Compared with our DFT selection, MP2 as a post-HF method is costlier than the pure functional BLYP, and def2-SVP is comparable to 6-31G* but with the RI approximation the former is much faster than the latter. Compared with the semi-empirical PM6-D3H4X, MP2/6-31G* it is costlier in any sense. For the large macrocycle containing ~150 atoms it is computationally demanding to do higher-level calculations at MP2/6-31G*. Thus, we grab a DFT selection with acceptable accuracy and affordable cost, i.e., BLYP-D4/def2-SVP. PM6-D3H4X as a semi-empirical QM method is itself of low accuracy, but we cannot ensure that using it would degrade the force-field accuracy as the refitting is performed in a molecule-specific manner. Compared with the transferable GAFF derivatives, we have more degrees of freedom in our refitting and thus the low-quality PM6-D3H4X level may not necessarily make the situation worse. Therefore, we performed force-field refitting and practically simulate the system and compute the binding information, in order to reach an ultimate conclusion about the QM-level selection and the force field performance.
- The authors used a contact number metric to describe the spatial relationshio between the host and guest and also mentioned that it is insufficient to differentiate different inter-molecular interaction patterns properly, thus they also used another contact number focusing on host-phenyl contacts. Did the authors try other simple metrics such as center-of-mass distance to complement the analysis? Distance between COMs can define how far away the two molecules are seperated while contact number tells the possible interaction density.For similar contact number, COM can help differentiate side interactions from in-cavity interactions.
Response: Before projecting the C-C_Ph free energy surface, we actually have already performed the COM-distance-C analyses suggested by the reviewer. The radius-contact free energy surfaces are presented in Fig. S9-S11, and related discussions are originally presented in section 3.3 but now have been moved to the supporting information in the revised manuscript.
- Pay attention to typos, e.g. “complexity of the host-host interactions”should be “complexity of the host-guest interactions”.
Response: Typo corrected. Thank you for your careful reading.

Reviewer 2 Report
A General Picture of Cucurbit[8]uril Host-Guest Binding: Recalibrating Bonded Interactions
Zhaoxi Sun1* , Qiaole He2 , Zhihao Gong3-4 , Payam Kalhor1 , Zhe Huai5* and Zhirong Liu1. 1College of Chemistry and Molecular Engineering, Peking University, Beijing 1008
The abstract provides a clear overview of the research, highlighting the difficulty of understanding host-guest interactions in supramolecular chemistry, specially on supramolecular containers of the Cucurbiturils family, which show promising drug carrying capabilities. The specific compound studied, Cucurbit[8]uril (CB8), is introduced along with its ability to exploit host-guest recognition motifs. This work extends previous work on the binding of seven structurally diverse drugs to CB8, commenting on the comparison of two fixed-charge models for drug-like molecules and iterative refitting of atomic charges which led to improved binding thermodynamics. The focus of this work is on the evaluation of the bonded interactions as given by the General Amber force field. As the interaction pattern and binding thermodynamics are dependent on the modeling parameters, the authors assess and refit the force fields to improve the intra-molecular conformational preference and the description of inter-molecular host-guest interactions. Combining the charge-scheme comparison and the force-field leads to significant improvement in consistency with regard to experimental reference data.
1. Introduction
…Among the CB[n] family, Cucurbit[8]uril (CB8 or CB[8], see Fig. 1a) has intermediate portal size and cavity volume, which enables it to exploit almost all host-guest recognition motifs formed by this host family….
Q. ¿What are the volume and opening size dimensions? As the guest molecules are so structurally different, it would be interesting to show the window size, for example, and comparatively show the size (Kinetic diameter, perhaps) of the guests.
…However, molecular dynamics (MD) simulations of biological molecules with modern computational resources accessible to ordinary researchers can only achieve μs, which is still far from the biologically relevant time scales…
Comment. Only to highlight the use of the term “ordinary researchers” in this paragraph, which quickly prompts the question: define non ordinary researchers, or perhaps, extra-ordinary researchers.
…Although the quantum mechanics (QM) treatment is accurate and transferrable, all-atom molecular mechanics (MM) force fields are widely employed for biological systems due to efficiency considerations and the size, complexity, and intrinsic time scale of the investigated problem…
Comment. Also, QM methods applied to large and intrinsically complex biological systems are well beyond the capabilities of modern computational tools (Hardware/Software) in terms of simulation times.
…Therefore, in this work, we focus on the assessment and recalibration of the intra-molecular interaction terms… / …refitting the bonded terms with the generalized force-matching (FM) method for the host and guest molecules under investigation…
Comment. For clarity, it would be good to mention the method (generalized force-matching (FM)) in the abstract. Only a suggestion.
…Combining the central results in the current bonded-term recalibration and our previous charge-scheme assessment, some useful guidelines for the modeling of CB8 host-guest systems could be summarized. More generally, the results are expected to be useful in the modeling of all host-guest complexes…
Comment/Question. When referring to “All Host-Guest complexes”, are you referring specifically to “All CB8 protonated and deprotonated host-guest systems”?
Comment. The introduction provides a nice and clear overview of previous research, the system under study and the methodology used for the work.
2. Methodology and Computational Details
The starting structure of each host-guest complex is obtained by simply superposing the center of masses (COM) of the host and that of the guest. Solvation is performed with TIP3P water and the minimum distance between the solute surface and the box edge is set to 25 Å, which is sufficiently large to define a decoupled state. The truncated octahedron cell is replicated in whole space with periodic boundary conditions. Non-polarizable monovalent spherical counter ions of Na+ or Cl- parameterized for TIP3P water are added for neutralization.
Comment. Details of the simulation cell conditions are clear and sound for the system under study, as well as the considerations for free energy calculations.
…To avoid overfitting and the existence of unphysical parameters, a weak harmonic (L2) regularization term is applied to restrain the parameter space to explore in the parameter adjustment. The relative weights of the energy and force terms are the same, and that of the regularization term is 0.1 of the other two terms, leading to an intermediate level of regularization…
Q. Would you consider it useful, and within the scope of this study, to use and compare larger/lower regularization values? Could you elaborate on the choice of intermediate levels of regularization?
…22000 structures are used to initiate QM calculations. We have tested that this sample size or sampling time is already sufficient to converge the parameter set. Further adding samples (i.e., lengthening sampling time) leads to negligible changes in the outcomes…
Q. Are there data results supporting this sample size testing step?
3. Result and discussion
3.1. Recalibrating the bonded interactions.
3.2. A closer view of different parameter sets.
3.3. Binding modes.
3.4. Binding thermodynamics.
3.5. Guidelines for Host-guest Modelling.
3.6. Further improvements.
Although Figures properly show the obtained data, and a thorough discussion of every aspect of each one is presented, it is somewhat cumbersome to follow such a lengthy and in-depth analysis. I suggest, in order to keep the attention of the reader, to avoid repetitive analysis.
Figure 2. / Figure 3. / Figure 5. / Figure 6. / Figure 7. / Figure 8.
Example: Figure 4 takes more than a page to show information regarding the the most significant difference between GAFF derivatives, superposition of the host configurations, from b) GAFF, c) GAFF2, d) the FM-PM6 parameters initiated from GAFF, e)FM-PM6 parameters initiated from GAFF2, and f) FM-BLYP initiated from GAFF2. g) The time series of the radius of gyration of the CB8 ring under the GAFF2 and the refitted parameter sets.
Most of the graphical information provided could be presented as supplementary information, and discussion should be addressed in a more general manner. If more detailed info is necessary, the authors should point to the supplementary information.
Example: Figure 9 nicely shows a Correlation between the binding affinities Vs experimental data, which shows the level of improvement of the refitted parameters of the GAFF.
I consider that tables, such as Table 2, summarizes most of the obtained data and should be the base for the discussion. I really couldn’t follow the whole results and discussion section because it addresses too many details and it becomes too repetitive.
Author Response
Submission ID: molecules-2241132
Responses to Referees’ comments
------------------------------------
COMMENTS TO AUTHOR:
Reviewer #2:
A General Picture of Cucurbit[8]uril Host-Guest Binding: Recalibrating Bonded Interactions
Zhaoxi Sun1* , Qiaole He2 , Zhihao Gong3-4 , Payam Kalhor1 , Zhe Huai5* and Zhirong Liu1. 1College of Chemistry and Molecular Engineering, Peking University, Beijing 1008
The abstract provides a clear overview of the research, highlighting the difficulty of understanding host-guest interactions in supramolecular chemistry, specially on supramolecular containers of the Cucurbiturils family, which show promising drug carrying capabilities. The specific compound studied, Cucurbit[8]uril (CB8), is introduced along with its ability to exploit host-guest recognition motifs. This work extends previous work on the binding of seven structurally diverse drugs to CB8, commenting on the comparison of two fixed-charge models for drug-like molecules and iterative refitting of atomic charges which led to improved binding thermodynamics. The focus of this work is on the evaluation of the bonded interactions as given by the General Amber force field. As the interaction pattern and binding thermodynamics are dependent on the modeling parameters, the authors assess and refit the force fields to improve the intra-molecular conformational preference and the description of inter-molecular host-guest interactions. Combining the charge-scheme comparison and the force-field leads to significant improvement in consistency with regard to experimental reference data.
- Introduction
…Among the CB[n] family, Cucurbit[8]uril (CB8 or CB[8], see Fig. 1a) has intermediate portal size and cavity volume, which enables it to exploit almost all host-guest recognition motifs formed by this host family….
- ¿What are the volume and opening size dimensions? As the guest molecules are so structurally different, it would be interesting to show the window size, for example, and comparatively show the size (Kinetic diameter, perhaps) of the guests.
Response: The portal diameters, the cavity diameter and many other properties of CBn have been reported in references. In the revision, we added some clarifications to the end of the mentioned sentence and cite related references containing these length parameters.
…However, molecular dynamics (MD) simulations of biological molecules with modern computational resources accessible to ordinary researchers can only achieve μs, which is still far from the biologically relevant time scales…
Comment. Only to highlight the use of the term “ordinary researchers” in this paragraph, which quickly prompts the question: define non ordinary researchers, or perhaps, extra-ordinary researchers.
Response: We remove the ‘ordinary researchers’ in this sentence in revision.
…Although the quantum mechanics (QM) treatment is accurate and transferrable, all-atom molecular mechanics (MM) force fields are widely employed for biological systems due to efficiency considerations and the size, complexity, and intrinsic time scale of the investigated problem…
Comment. Also, QM methods applied to large and intrinsically complex biological systems are well beyond the capabilities of modern computational tools (Hardware/Software) in terms of simulation times.
Response: This hardware/software point has been added to this sentence in revision.
…Therefore, in this work, we focus on the assessment and recalibration of the intra-molecular interaction terms… / …refitting the bonded terms with the generalized force-matching (FM) method for the host and guest molecules under investigation…
Comment. For clarity, it would be good to mention the method (generalized force-matching (FM)) in the abstract. Only a suggestion.
Response: The generalized force-matching keyword has been added to the abstract in revision.
…Combining the central results in the current bonded-term recalibration and our previous charge-scheme assessment, some useful guidelines for the modeling of CB8 host-guest systems could be summarized. More generally, the results are expected to be useful in the modeling of all host-guest complexes…
Comment/Question. When referring to “All Host-Guest complexes”, are you referring specifically to “All CB8 protonated and deprotonated host-guest systems”?
Response: The sampling protocol and force-field parametrization (charge scheme in the previous work and bonded parameters refitting in the current paper) offer a general route to access host-guest binding modes and affinities. Therefore, all host-guest complexes means all the pairs of macrocyclic receptors and drug-like small molecules. Note that similar protocols have also been applied to cyclodextrins in another paper published by us recently, Carbohydrate Polymers 297, 120050. We have also successfully applied the modelling protocol in ongoing projects investigating other host-guest complexes (paper to be submitted).
Comment. The introduction provides a nice and clear overview of previous research, the system under study and the methodology used for the work.
Response: Thank you for your reading and admiring the introduction section.
- Methodology and Computational Details
The starting structure of each host-guest complex is obtained by simply superposing the center of masses (COM) of the host and that of the guest. Solvation is performed with TIP3P water and the minimum distance between the solute surface and the box edge is set to 25 Å, which is sufficiently large to define a decoupled state. The truncated octahedron cell is replicated in whole space with periodic boundary conditions. Non-polarizable monovalent spherical counter ions of Na+ or Cl- parameterized for TIP3P water are added for neutralization.
Comment. Details of the simulation cell conditions are clear and sound for the system under study, as well as the considerations for free energy calculations.
Response: Great.
…To avoid overfitting and the existence of unphysical parameters, a weak harmonic (L2) regularization term is applied to restrain the parameter space to explore in the parameter adjustment. The relative weights of the energy and force terms are the same, and that of the regularization term is 0.1 of the other two terms, leading to an intermediate level of regularization…
- Would you consider it useful, and within the scope of this study, to use and compare larger/lower regularization values? Could you elaborate on the choice of intermediate levels of regularization?
Response: The strength of the L2 restraint depends on how much we trust the pre-fitted GAFF derivatives (initial guess). Concerning the strength of regularization discussed here, we use the relative weights of the energy, force and regularization terms and explicitly define the 1:1:0.1 ratio as the intermediate level, which is an empirical rule according to our experiences. If we set the weights of different terms to the same, the regularization strength is often large/stiff, while relative weights such as 1:1:0.01 are considered weakly regularized. Aside from the weights of different terms, the regularization strength could also be adjusted by varying the other terms in the regularization term, e.g., the width of the prior distribution of parameters. These points are rather technical and are solely force-field parametrization basics that are difficult to be fully explained in this paper. Thus, we would recommend readers to read related references e.g., the original paper of the generalized force-matching (doi: 10.1021/acs.jcim.0c01444) for details.
…22000 structures are used to initiate QM calculations. We have tested that this sample size or sampling time is already sufficient to converge the parameter set. Further adding samples (i.e., lengthening sampling time) leads to negligible changes in the outcomes…
- Are there data results supporting this sample size testing step?
Response: Yes, there are some quantitative metrics to check for this refitting convergence. Normally, we can use the parameter variations when further samples are added to the training set to check convergence of the refitting procedure, and we indeed do extensive tests for the PM6-D3H4X set. We initially gathered 24000 configurations at this PM6-D3H4X reference level, divide the training set into pieces and gradually add new samples to the training set. Below are the variations (RMSD) of all parameters during the sample-size addition. We can see that the parameter adjustment already converges very well with 20000 samples. For safety and avoid wasting existing results, we thus pick the 22000 as the sample size used in this paper. As these results are rather technical, we do not include these data in the article.
|
Sample size |
parameter RMSD |
|
1200 |
0.08174080232500464 |
|
2400 |
0.03635879358180192 |
|
3600 |
0.08187789008244514 |
|
4800 |
0.014641107167176511 |
|
6000 |
0.012974578813000733 |
|
7200 |
0.004313087547745715 |
|
8400 |
0.003913153351825079 |
|
9600 |
0.0033754564291740134 |
|
10800 |
0.00013595083750493045 |
|
12000 |
0.0048085193855929844 |
|
13200 |
0.00013093484403782634 |
|
14400 |
6.613017800701132e-05 |
|
15600 |
0.006532645843002652 |
|
16800 |
7.820828538850201e-05 |
|
18000 |
0.004404538264374155 |
|
19200 |
6.301400824851158e-05 |
|
20400 |
5.716495706126166e-05 |
|
21600 |
5.695130586892616e-05 |
|
22800 |
7.901641729289348e-05 |
|
24000 |
2.0558560758786707e-05 |
- Result and discussion
3.1. Recalibrating the bonded interactions.
3.2. A closer view of different parameter sets.
3.3. Binding modes.
3.4. Binding thermodynamics.
3.5. Guidelines for Host-guest Modelling.
3.6. Further improvements.
Although Figures properly show the obtained data, and a thorough discussion of every aspect of each one is presented, it is somewhat cumbersome to follow such a lengthy and in-depth analysis. I suggest, in order to keep the attention of the reader, to avoid repetitive analysis.
Figure 2. / Figure 3. / Figure 5. / Figure 6. / Figure 7. / Figure 8.
Example: Figure 4 takes more than a page to show information regarding the most significant difference between GAFF derivatives, superposition of the host configurations, from b) GAFF, c) GAFF2, d) the FM-PM6 parameters initiated from GAFF, e)FM-PM6 parameters initiated from GAFF2, and f) FM-BLYP initiated from GAFF2. g) The time series of the radius of gyration of the CB8 ring under the GAFF2 and the refitted parameter sets.
Most of the graphical information provided could be presented as supplementary information, and discussion should be addressed in a more general manner. If more detailed info is necessary, the authors should point to the supplementary information.
Example: Figure 9 nicely shows a Correlation between the binding affinities Vs experimental data, which shows the level of improvement of the refitted parameters of the GAFF.
I consider that tables, such as Table 2, summarizes most of the obtained data and should be the base for the discussion. I really couldn’t follow the whole results and discussion section because it addresses too many details and it becomes too repetitive.
Response: The long discussions about the radius-contact surfaces have been moved to the supporting information. Many parts of the main text (section 2, 3.1, 3.2 and 3.3) have been shortened. As a result, the whole main text has been shortened by 6 pages.
Fig. 2-3 present the energetic and force errors for evaluation and Fig. 4 presents the torsional term different in GAFF and GAFF2 and the host dynamics produced by different parameter sets, all of which are actually the core features of the current ‘bonded term evaluation and refitting’ paper. Further, as most figures containing related data but in greater detail are already included in the supporting information (e.g., energetic and force errors in Fig. S1-S4, term-specific illustration of GAFF torsions in Fig. S5, fluctuations of host-guest contacts in Fig. S6-S8), we already distil the core and crucial figures to be included in the main article as much as we can.

Round 2
Reviewer 2 Report
Comments, suggestions and corrections have been addressed in an appropriate and timely manner. A minor suggestions on Figure 5: Cyan and magenta oval highlighters cover the graph title and legend box text.
Excellent work, looking forward to see it published.